# A solid-state lithium-ion battery with micron-sized silicon anode operating free from external pressure

Hui Pan [1], Lei Wang[1], Yu Shi[1], Chuanchao Sheng[1], Sixie Yang[2], Ping He [1] ✉ & Haoshen Zhou [1] ✉

Applying high stack pressure (often up to tens of megapascals) to solid-state Li-ion batteries is primarily done to address the issues of internal voids formation and subsequent Li-ion transport blockage within the solid electrode due to volume changes. Whereas, redundant pressurizing devices lower the energy density of batteries and raise the cost. Herein, a mechanical optimization strategy involving elastic electrolyte is proposed for SSBs operating without external pressurizing, but relying solely on the built-in pressure of cells. We combine soft-rigid dual monomer copolymer with deep eutectic mixture to design an elastic solid electrolyte, which exhibits not only high stretchability and deformation recovery capability but also high room-temperature Li-ion conductivity of $2 \times 10^{-3}$ S cm$^{-1}$ and nonflammability. The micron-sized Si anode without additional stack pressure, paired with the elastic electrolyte, exhibits exceptional stability for 300 cycles with 90.8% capacity retention. Furthermore, the solid Li/elastic electrolyte/LiFePO$_4$ battery delivers 143.3 mAh g$^{-1}$ after 400 cycles. Finally, the micron-sized Si/elastic electrolyte/LiFePO$_4$ full cell operates stably for 100 cycles in the absence of any additional pressure, maintaining a capacity retention rate of 98.3%. This significantly advances the practical applications of solid-state batteries.

As the grid-scale energy storage market continues to prosper, conventional Li-ion batteries with organic liquid electrolytes are failing to meet the increasingly urgent demands for high energy density and safety. The development of solid-state batteries represents a fundamental solution to these challenges, primarily due to the intrinsically higher safety offered by solid-state electrolytes (SSEs) and the potential utilization of electrode materials with high specific capacity, such as Li and Li-based alloys[1,2]. Several successful attempts have been made in constructing solid-state batteries with excellent cycle stability[3–8]. However, the practical application of solid-state batteries is hindered by certain technical issues, including the high stack pressure required for these batteries.

To date, the requirement of high stack pressure, reaching up to several tens or even several hundred megapascals (MPa), remains essential for the long-cycle stability of solid-state batteries employing inorganic SSEs, both during their fabrication and operation. For instance, recent studies on Li-In||Ni$_{0.83}$Co$_{0.11}$Mn$_{0.06}$O$_2$ (NCM-83)[9] and Li-In||LiCoO$_2$ (LCO)[10] batteries utilizing oxychloride SSEs have demonstrated exceptional cycle stability under stack pressures of 80 MPa and 190 MPa, respectively. Despite the impressive electrochemical performance of solid-state batteries, providing such high stack pressure in practical applications poses significant challenges. Moreover, the inclusion of additional pressurizing equipment substantially diminishes the energy density of the battery and increases

[1]Center of Energy Storage Materials & Technology, College of Engineering and Applied Sciences, Jiangsu Key Laboratory of Artificial Functional Materials, National Laboratory of Solid State Microstructures and Collaborative Innovation Center of Advanced Microstructures, Nanjing University, Nanjing 210093, P. R. China. [2]School of Materials Science and Intelligent Engineering, Nanjing University, Suzhou 215163, P. R. China. ✉e-mail: pinghe@nju.edu.cn; hszhou@nju.edu.cn

production costs. Reducing the operating pressure often results in decreased cycling stability for solid-state batteries. For instance, a Li-In/Li$_3$InCl$_6$/NCM-83 cell operated under 2 MPa experienced a rapid capacity degradation, retaining only 65% of its capacity after 50 cycles[11]. This work utilized Li-In alloy as the negative electrode addressing the incompatibility issues between the electrolyte and metallic Li. However, the battery still faced relatively rapid degradation, which was attributed to void formation in the electrode and loss of contact between SSEs and NCM-83. The necessity of high stack pressure remains a significant impediment to the practical implementation of solid-state batteries.

The primary factors contributing to this dilemma is the inflexibility of inorganic SSEs combined with the volume changes experienced by the active electrode materials during cycling. Typically, cathode materials, such as LCO, LiFePO$_4$ (LFP), and nickel-rich lithium transition metal oxides, exhibit volume change rates ranging from 2% to 7% during lithiation and delithiation[12]. However, the situation becomes even more challenging when considering anode materials, particularly those with high specific capacities, which can undergo more significant volume fluctuations during cycling. In an extreme case, the volumetric expansion of silicon (Si) during lithiation can reach up to 400%[13]. Consequently, solid-state cells incorporating Si anodes necessitate high stack pressures of 50–150 MPa to maintain a mechanically functional Si/electrolyte interface[14–16]. Without sufficient external pressure, the repeated expansion and contraction of the active materials can lead to the loosening of the contact between the

active materials and SSEs. This, in turn, hinders Li-ion transport and can result in the disintegration of the electrode (Fig. 1a, b). While applying substantial pressure can certainly enhance the performance of solid-state batteries (Fig. 1c), achieving such rigorous conditions in practical applications is a formidable challenge. Finding ways to achieve high stability in solid-state batteries under low stack pressure, or even refraining from external pressure, is currently an urgent problem that needs to be addressed.

As illustrated in Fig. 1d, e, we present a design concept to tackle the previously mentioned troubles. By introducing an elastic solid electrolyte into the porous electrode, one that features high ionic conductivity, substantial tensile and compressive resilience, non-combustibility, and even self-healing capability, we propose a potential solution to the stress failure issues of electrode. We envision this versatile elastic solid electrolyte to permeate and occupy the internal voids of the porous electrode, and have the ability to encapsulate the irregularly shaped active material particles within the electrode structure (as depicted in Fig. 1d). In this configuration, the solid electrolyte functions as a rapid ion transport channel between active material particles (see the magnified view in Fig. 1d). As electrode reactions progress, the active material particles may undergo cracking due to substantial volume changes (as depicted in Fig. 1e). Nevertheless, previous works have validated the effectiveness of using elastic media to alleviate the Si electrode degradation in the cells with liquid electrolytes[17,18]. Therefore, it is reasonable to speculate that even though the elastic solid electrolyte cannot completely prevent such deformations and fractures, it can effectively encase the active

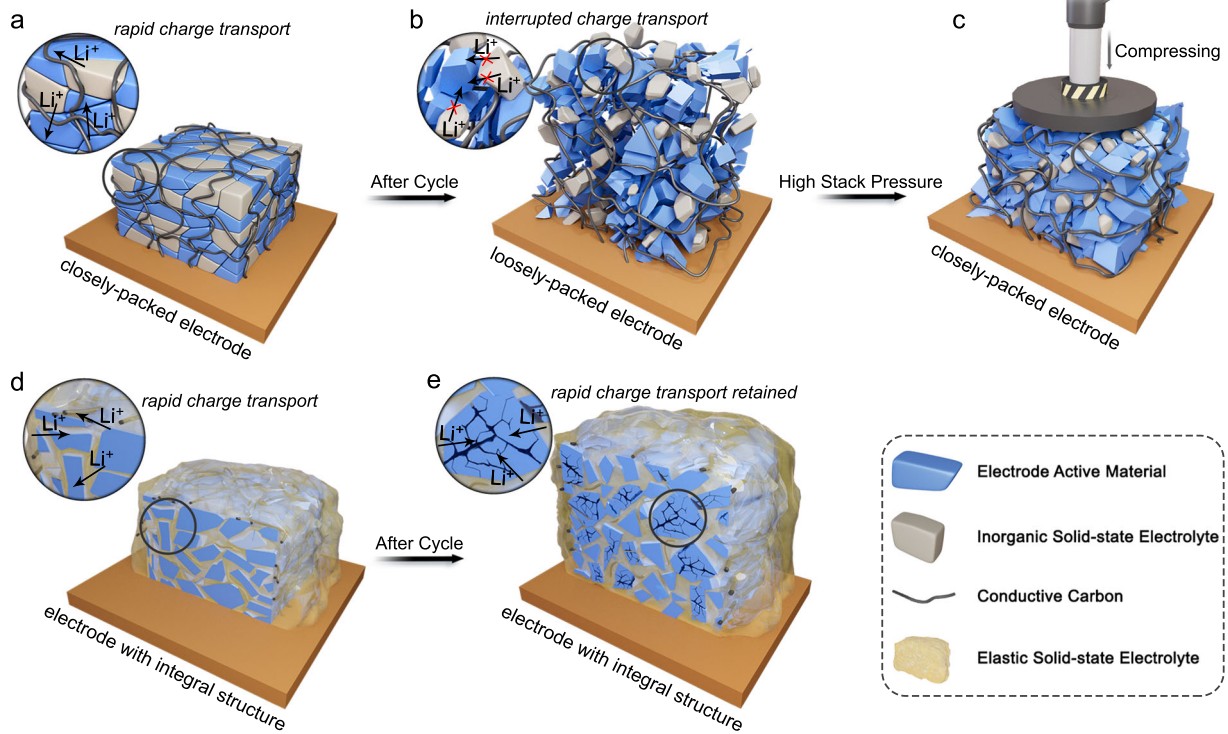

**Fig. 1 | Schematics of the failure mechanism of the solid-state electrode and the solutions. a** The solid-state electrode with the inorganic solid-state electrolyte (**b**) undergoes pulverization after cycles owing to the large volume change of the electrode active materials. **c** The application of high stack pressure may relieve the deterioration of electrode, but the redundant pressurizing devices greatly reduce the energy density of the battery and raise the cost. **d** The elastic electrolyte with high ionic conductivity fills the porous electrode and surrounds the active materials. **e** Due to the excellent tensile and compressive elasticity of the electrolyte, the active materials cannot break free from the elastic electrolyte during lithiation and delithiation, despite of the large volume change. In this way, unimpeded Li-ion transport inside the electrode is realized.

material when volume changes occur, thereby maintaining the efficient Li-ion transport.

Most recently, a viscoelastic glass electrolyte LiAlCl$_4$−75%O (LACO75) with benign flexibility has been developed address the contact loss of the cathode active materials and the electrolyte under low stack pressure[19]. The proposed Li/Li$_{6.4}$La$_3$Zr$_{1.4}$Ta$_{0.6}$O$_{12}$/LACO75-LiNi$_{0.6}$Co$_{0.2}$Mn$_{0.2}$O$_2$ cell functioned well under kilopascal pressure, partially validating the design concept involving the elastic electrolyte to mitigate volume change. However, it's worth noting that the viscoelastic electrolyte in this work demonstrated compatibility exclusively with the cathode materials. Nevertheless, the stress-related failure issue becomes significantly more serious in the anode due to the more drastic volume change of the anode active materials. It is uncertain whether the incompatibility of LACO75 to the anode was caused by its limited electrochemical window or the excessive volume fluctuation of the anode. Additional concern is that the cell was operated at elevated temperature (60 °C), which limits its applicability. To realize the aforementioned design concept, an alternative worth considering is the use of solid polymer electrolytes. However, the requirements of this design cannot be met by conventional polymer electrolytes. Polymer electrolytes previously reported either exhibit low room-temperature conductivity[20], poor tensile elasticity, or require liquid organic electrolytes[21] which leads to safety concerns. More importantly, they are unable to infiltrate the internal structure of porous electrodes and achieve a close integration with the active material particles within the electrode.

A copolymer material consists of soft and rigid phases has been reported to exhibit outstanding mechanical properties including high strength, good deformation resiliency, and self-healing property[22]. Provided that this copolymer is endowed with high ionic conductivity without scarifying mechanical properties, it could be an ideal electrolyte to meet our demand. Inspired by this, an elastic solid electrolyte comprising of copolymer matrix and deep eutectic mixture (DEM) showing high room-temperature ionic conductivity of $2 \times 10^{-3}\,S\,cm^{-1}$, excellent stretchability, high fracture strength, desirable fatigue resistance, and nonflammability is devised in this work. Micron-sized Si (μm-Si) electrode with the most significant volume change was selected as the electrode active material. Moreover, a stress monitoring technique using a home-made pressure sensing unit was employed to measure the internal pressure of the assembled cells. As a result, the solid-state μm-Si electrode with the elastic electrolyte delivered outstanding cycle stability under 546 kPa, which was the built-in pressure of the coin-type cell in the absence of external pressurizing device. Coupling with the well-performing LFP cathode, the μm-Si/elastic electrolyte/LFP pouch cell functioned normally without external stack pressure and withstood the cutting and bending conditions, exhibiting distinguished security and promising application prospects. This work realizes the safe and stable operation of the solid-state full batteries free from additional pressurization, thereby greatly promotes their practical process.

## Results and discussion
### Synthesis and physicochemical characterization of the elastic electrolyte

The elastic electrolyte was synthesized through UV polymerization of dimethyl acrylamide (DMAM) and acrylamide (AM) in DEM with different monomer ratios. DEMs are the eutectic mixtures of two solid chemicals with strong interaction, which have much lower melting point than the individual component. DEMs emerge as the electrolytes for Li-ion batteries[23], Li-oxygen battery[24], and organic batteries[25] owing to their high ionic conductivity, non-toxic and environmental friendliness[26]. Moreover, DEMs surpass conventional organic liquid electrolytes in safety because of its incombustibility and low vapor pressure[27], making it an ideal choice to fabricate safe electrolyte. DEM used in this work specifically refers to the eutectic mixture of solid *N*-

methylacetamide (NMA) and lithium bis-fluorosulfonimide (LiFSI) in the molar ratio of 4:1. The detailed discussion about the selection and properties of the DEMs can be found in the Supplementary Note.

As shown in Fig. 2a, poly-DMAM is miscible with DEM and forms homogeneous and transparent soft phase (Supplementary Fig. 4, $x = 0$, $x$ represents the proportion of poly-AM to the whole polymer); while poly-AM is immiscible with DEM and contains abundant hydrogen bonds, making poly-AM the rigid phase with white color (Supplementary Fig. 4, $x = 1$). The rigid and soft phase copolymerize randomly in the electrolyte and a bicontinuous phase-separate network forms due to the disparate miscibility of poly-DMAM and poly-AM with DEM. The opacity of the copolymer increased with the ratio of poly-AM, with a transition point at $x = 0.8$ (Supplementary Figs. 4 and 5). It is systematically studied how the proportion of the soft and rigid phase impacts the morphology, microstructure and glass transition temperature ($T_g$) of the copolymer. The microscopic morphology of the polymers was characterized using scanning electron microscopy (SEM) and atomic force microscopy (AFM). It can be observed that the surface of pure poly-DMAM was flat and smooth (Fig. 1b, $x = 0$) with an average roughness of 2 nm (Supplementary Fig. 6a). Granular morphology was observable when $x = 0.5$ due to the introduction of rigid phase. The surface fluctuations enhanced with the increase of rigid phase, and the average roughness reached 145 nm in pure poly-AM (Supplementary Fig. 6d). Small-angle X-ray scattering (SAXS) was used to study the phase separation of the copolymer (Supplementary Fig. 7). Pure poly-AM and poly-DMAM showed no obvious scattering peak, indicating that they were homogenous polymer without phase separation. Whereas the copolymer ($x = 0.8$) showed a single broad scattering peak at $0.01-0.03\,Å^{-1}$, suggesting a microphase separation with domain size of 20.9−62.8 nm. In addition to the morphology and microstructure, the different ratio of rigid and soft phase has an impact on the $T_g$ of the copolymer. As demonstrated by the differential scanning calorimetry measurement results in Supplementary Fig. 8, $T_g$ declined with the increased ratio of poly-AM. All of the samples have $T_g$ lower than −60 °C, enabling them with excellent ion transport ability at room temperature, which will be discussed later. After integrating with the copolymer network, the DEM is constrained and coordinated by the large molecular chains inside the polymer. As shown in Supplementary Fig. 9, the evaporation temperature of NMA increases from 82.2 °C in free-state DME to 204.6 °C in the copolymer. Besides, decomposition temperature of LiFSI salt rises from 255 to 325 °C. The significant increasements in transition temperatures can be attributed to the effects of confinement and coordination of the DEM inside the copolymer framework, which makes it distinct from conventional free-state liquid additives, and exhibit a tendency towards solid-state characteristics.

### Mechanical properties of the elastic electrolyte

As aforementioned, the mechanical properties of the electrolyte have great impact on the cycle stability of solid-state electrode, especially under low stack pressure. To optimize the mechanical properties of the solid electrolyte, uniaxial tensile tests were conducted on copolymers with different proportions of poly-DMAM (soft phase) and poly-AM (rigid phase). As shown in Fig. 2c, pure poly-DMAM exhibited outstanding stretchability with a large break elongation of over 1400%, whereas it had a low fracture strength of only 0.08 MPa. The low fracture strength was caused by the lack of interchain interaction, as the poly-DMAM was miscible with DEM and the molecular chains were highly solvated. In contrast, pure poly-AM presented negligible elongation before break and showed high stiffness. Combining the stretchability of soft phase and the strength of rigid phase, the copolymer with $x = 0.8$ displayed reinforced mechanical properties including large break elongation of 1160% and high fracture strength of 1.7 MPa. Moreover, the copolymer had excellent shape memory ability and demonstrated quick deformation recovery within 5 min after

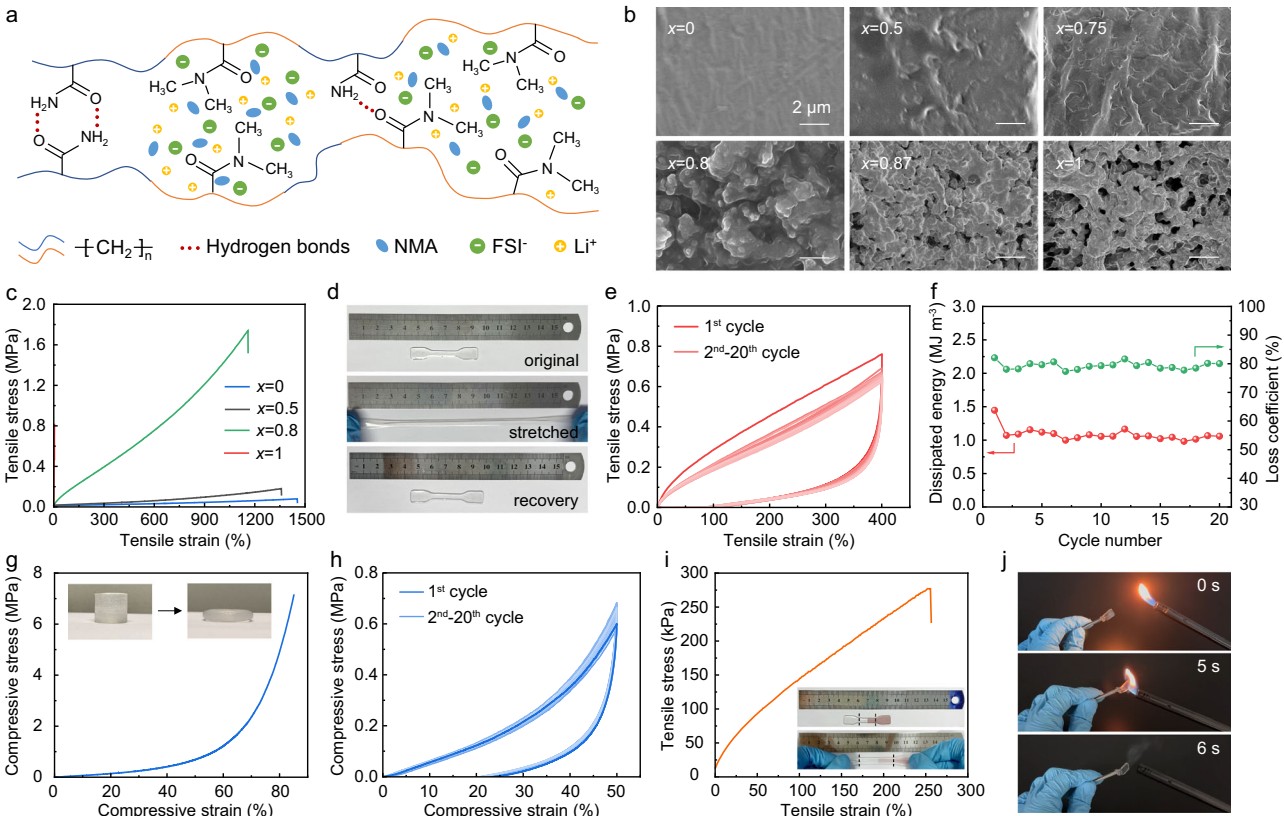

**Fig. 2 | Characteristics and mechanical properties of the elastic electrolyte.** **a** Schematic of the molecular design of the elastic electrolyte. **b** SEM images of the electrolyte with different ratios of poly-AM (wherein $x$ refers to the ratio of poly-AM). **c** Tensile stress-strain curves of the elastic electrolyte with different ratios of poly-AM. **d** Digital images showing the stretchability and deformation recovery of the elastic electrolyte with $x = 0.8$. **e** Cyclic tensile stress-strain curves with the elongation of 400% of the elastic electrolyte with $x = 0.8$ and (**f**) the corresponding energy dissipation for each cycle. **g** Compressive stress-strain curve with the compression ratio of 85% and (**h**) cyclic compressive stress-strain curves with the compression ratio of 50% of the elastic electrolyte with $x = 0.8$. The test sample rested for 5 min between each cycle during the cyclic tests for the reconstruction of hydrogen bonds. **i** The tensile stress-strain curve of the self-healing elastic electrolyte sample. Source data are provided as a Source Data file. **j** Digital images of the combustion tests of the elastic electrolyte.

stretched to 450% in length (Supplementary Movie 1 and Fig. 2d). The cyclic loading-unloading tests of the copolymer with $x = 0.8$ was performed to assess the fatigue resistance of the electrolyte. The strain-stress curves were typical hysteresis loops with residual strain (Fig. 2e). The emergence of hysteresis loops was a result of the fracture and restoration of noncovalent bonds[28], namely hydrogen bonds in this case[29]. The break and reconstruction of hydrogen bonds will dissipate massive energy, which is anticipated to eliminate the energy yielded from the volume change of the electrode active materials and prevent the disintegration of the electrode[30]. After 5 min relaxation, the sample recovered to the initial length and exhibited excellent mechanical stability during the subsequent loading-unloading tests. The initial energy dissipation (the integrated area of the hysteresis loop) was 1.4 MJ m$^{-3}$ with a loss coefficient (the ratio of dissipated energy to input energy) of 82% in the condition of 400% tensile strain. During the sequent cycles, the energy dissipation remained around 1 MJ m$^{-3}$ with loss coefficient of ~80%, demonstrating great fatigue resistance of the phase-separate copolymer (Fig. 2f).

Given that the electrode active materials may also expand during cycle, the electrolyte is expected to undergo compression in this situation. Therefore, compressive stress-strain of the copolymer was examined. No indication of failure was detected even with a high compression strain of 85% (Fig. 2g). In addition, the copolymer also displayed excellent compressive recoverability, since the stress-strain curves overlapped with each other during 20 cycles with a compressive strain of 50% (Fig. 2h). The energy dissipation phenomenon was analogous to that in tensile tests, as illustrated by the hysteresis loops

in Fig. 2h and the energy dissipation curve in Supplementary Fig. 10. On account of the abundant noncovalent bonds between molecular chains, the copolymer manifested self-healing property. Two dumbbell-shaped samples with different colors were cut into half and then adhered to each other at 60 °C. After resting for 1 min, a healed sample was obtained (Supplementary Fig. 11) with a break elongation of over 250% and a fracture strength of 277 kPa (Fig. 2i). In addition, thanks to the flame retardance of the DEM, the elastic electrolyte exhibited outstanding nonflammability in the combustion tests (Fig. 2j). The superior reliability of the elastic electrolyte ensured the security of the battery.

To sum up, when $x = 0.8$, the copolymer combines the stretchability of soft phase and the strength of rigid phase and exhibited outstanding mechanical properties including large elongation of 1160% and facture strength of 1.7 MPa. The cyclic loading-unloading tests proclaimed excellent shape memory and fatigue resistance of the copolymer. Moreover, it displayed energy dissipation characteristics, which is beneficial to eliminate the energy imposed by the volume change of the active materials and maintain the integrity of the electrode. The outstanding mechanical properties originated from the microphase separation of the copolymer. To be more specific, the AM-rich segments contained abundant hydrogen bonds (as shown in Fig. 2a) which contributed to high strength and energy dissipation property; while the DMAM-rich segments featured weak interchain interactions and thus induce high deformability. Besides, the self-healing and nonflammability properties of the elastic electrolyte are strong guarantees of the safety of the battery. Based on the above

favorable features, the elastic copolymer is expected to be a superb electrolyte for the solid-state batteries working under low stack pressure.

## Ion transport properties and electrochemical stability of the elastic electrolyte

Typically, it is difficult for solid-state electrolyte to reach a counterbalance between high ionic conductivity and satisfying mechanical properties. After mechanical characteristics evaluation, the ion transport characters of the elastic polymers were studied to evaluate their feasibility as electrolytes. Firstly, the influence of the ratio of soft and rigid phases on ionic conductivity were measured. As shown in Supplementary Fig. 12, the variation of the ionic conductivity with $x$ was insignificant. All samples with the ratio of poly-AM from 0 to 0.87 had high ionic conductivities around $2 \times 10^{-3}$ S cm$^{-1}$. Given that the copolymer with $x = 0.8$ had the best mechanical properties and high ionic conductivity, this ratio was identified as the optimal choice for the elastic electrolyte and all studies henceforth were carried out based on it. Temperature-dependent ionic conductivity of the elastic electrolyte followed the Vogel-Tammann-Fulcher (VTF) equation (Fig. 3a), given by

$$\sigma(T) = AT^{-\frac{1}{2}} \exp\left[-\frac{E_a}{k_B(T - T_0)}\right] \quad (1)$$

where $A$ is the pre-exponential factor, $E_a$ is the pseudoactivation energy, $k_B$ is the Boltzmann constant, and $T_0$ is the equilibrium Vogel scaling temperature. The fitting parameters are summarized in Supplementary Table 1. The VTF behavior indicates that the ion transport was closely related to the molecular chain dynamics, which is typical for a polymer electrolyte[31,32]. On account of the low $T_g$ of the copolymer (−68.5 °C

when $x = 0.8$), the electrolyte displayed outstanding ionic conductivity of $2 \times 10^{-3}$ and $2 \times 10^{-4}$ S cm$^{-1}$ at ambient temperature and at low temperature of −10 °C, respectively (Fig. 3a and Supplementary Fig. 13). The Li$^+$ transference number ($t_{Li+}$) was calculated to be 0.44 (Fig. 3b), manifesting the superior Li$^+$ transport ability of the elastic electrolyte. The quick ion transport was originated from the dissociation of lithium salt. The fitting results of the Raman peak corresponding to S-N-S bending in the LiFSI (Fig. 3c) showed intense signal of free FSI- anion (731 cm$^{-1}$) and weak signal of contact ion pair (CIP, 744 cm$^{-1}$, one FSI$^-$ bonding with one Li$^+$)[33], certifying that the dissociation of LiFSI in the DEM was unaffected after uniting with the copolymer.

The electrochemical stability of the elastic electrolyte was assessed through galvanostatic Li plating and stripping tests and linear sweep voltammetry (LSV). The symmetric Li/elastic electrolyte/Li cell showed a low initial overpotential of 65 mV at 0.1 mA cm$^{-2}$, 0.1 mAh cm$^{-2}$ and cycled stably for 1800 h (Fig. 3d). In addition, the symmetric cell realized a stable operation for over 350 h at 0.2 mA cm$^{-2}$ and exhibited no failure under a high current density of 1 mA cm$^{-2}$ (Supplementary Fig. 14). As exhibited in Supplementary Fig. 15, the impedance of the symmetric cell increased within the first 4 h during the rest process due to SEI formed by the chemical reaction between the electrolyte and Li. But the increase in the impedance significantly slowed down after 4 h, suggesting the self-limitation of the chemical reaction. The electrochemical Li plating and stripping induced the further growth of SEI and the increment of the impedance. Fortunately, SEI between Li and the elastic electrolyte became stable after 50 h, and contributed to the subsequent long cycle stability of the symmetric cell. LSV tests were conducted to measure the oxidation potential of the electrolyte. As demonstrated by Fig. 3e, the onset oxidation potential for the elastic electrolyte after copolymerization

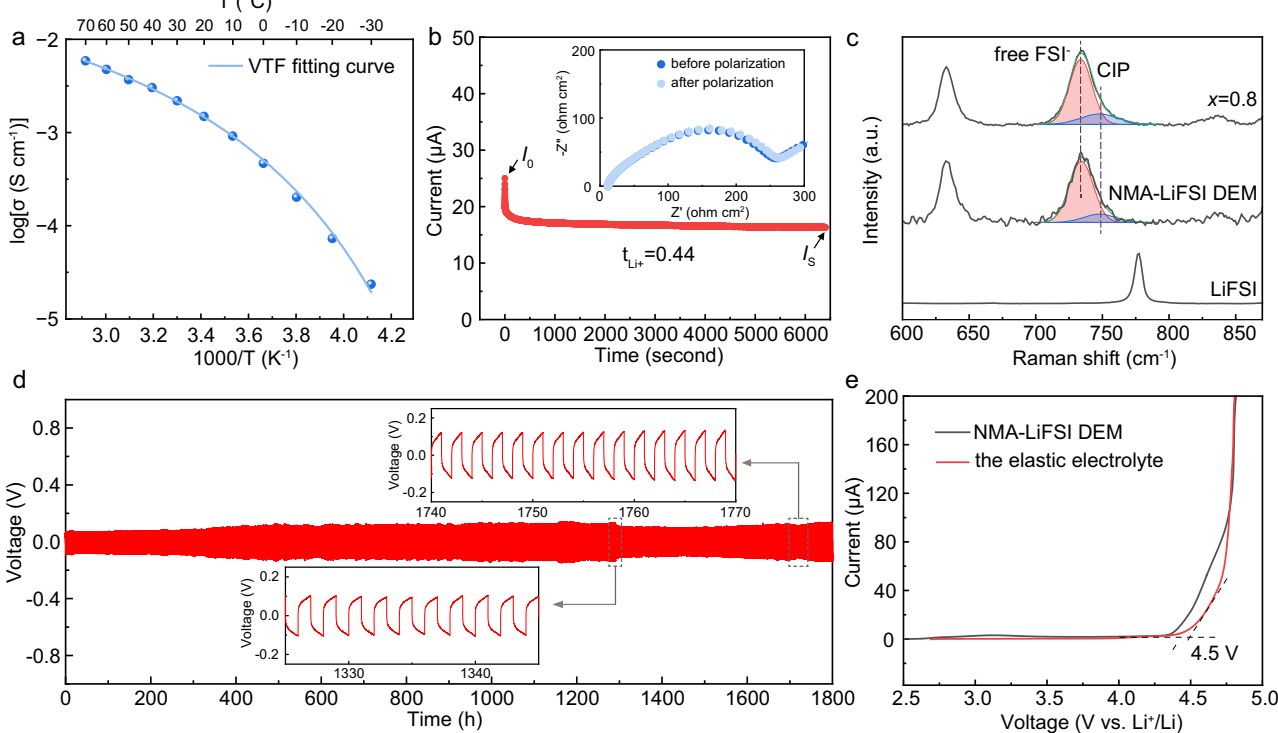

**Fig. 3 | Ion transport properties and the electrochemical stability of the elastic electrolyte. a** Temperature-dependent ionic conductivity of the elastic electrolyte and the corresponding Vogel-Tammann-Fulcher fitting curve. **b** Chronoamperometry curve of the Li/elastic electrolyte/Li symmetric cell with an applied voltage of 10 mV. Nyquist plots before and after polarization are inserted in the top right-hand corner. **c** Raman spectra of LiFSI, NMA-

LiFSI DEM, and the elastic electrolyte with $x = 0.8$. **d** Galvanostatic Li plating and stripping profiles of the Li/elastic electrolyte/Li symmetric cell at 0.1 mA cm$^{-2}$, 0.1 mAh cm$^{-2}$. **e** Linear sweep voltammetry curve of the NMA-LiFSI DEM and the elastic electrolyte at a sweep rate of 0.1 mV s$^{-1}$. Source data are provided as a Source Data file.

was around 4.5 V versus Li⁺/Li, which was a little bit higher than DEM. This oxidation potential is higher than the working potential of commonly used cathode materials including LFP, LCO and nickel-rich lithium transition metal oxides, ensuring the good compatibility between the electrolyte and the cathode.

In brief, the elastic electrolyte possessed a high ionic conductivity of $2 \times 10^{-3}$ S cm⁻¹ with $t_{Li+}$ of 0.44 at ambient temperature. The outstanding ion transport ability resulted from the considerable dissociation of lithium salt. Furthermore, the electrolyte exhibited excellent compatibility toward Li and a high onset oxidation potential of 4.5 V versus Li⁺/Li, suggesting its high electrochemical stability.

### Built-in pressure monitoring in cells and the fabrication of µm-Si anode with elastic electrolyte

Although no additional pressurization device is used in this work, there is still built-in pressure inside batteries. In order to study the effect of pressure on the solid-state electrode, the inner pressures of the coin-type and pouch-type batteries were measured through a home-made pressure sensing unit with a membrane force-sensitive resistance (MFSR). The commonly used load cell is too bulky to be placed inside the battery, thus a MFSR with a width of 14 mm and thickness of 0.2 mm was alternatively adopted. Firstly, the algebraic relationship

between the resistance of the MFSR and the applied force was determined. As shown in Fig. 4a, the applied force on the MFSR was calibrated through a load cell, while the resistance of the MFSR could be read directly on the multimeter. In this way, the response of resistance to force of the MFSR and the fitted standard response curve was obtained in Fig. 4b and c, respectively. Then, the calibrated MFSR was employed to measure the built-in pressure of the cells. As illustrated in Fig. 4d, e, the MFSR was placed between the cathode and anode (in the pouch-type cell) and two stainless steel discs (in the coin-type cell), with its two pins stuck out and connected to a digital multimeter. The resistance of the MFSR was shown on the multimeter, and the pressure inside the cells could be calculated thereby from the standard curve (Fig. 4c). As shown in Fig. 4f, the resistance of the sensor was 0.288 MΩ, corresponding to the built-in pressure of 546 kPa in the coin-type cell. The inner pressure of the pouch cell was determined to be 52 kPa with the similar method (Supplementary Fig. 16). It can be seen that the internal pressures of the coin-type pouch-type cells are much lower than that usually applied to solid-state batteries (at least a few MPa), and are readily achievable during the packaging of the batteries.

In light of these conditions, the effectiveness of employing the elastic electrolyte was assessed in both coin-type and pouch-type cells with µm-Si electrodes. Si is regarded as a competitive next-generation

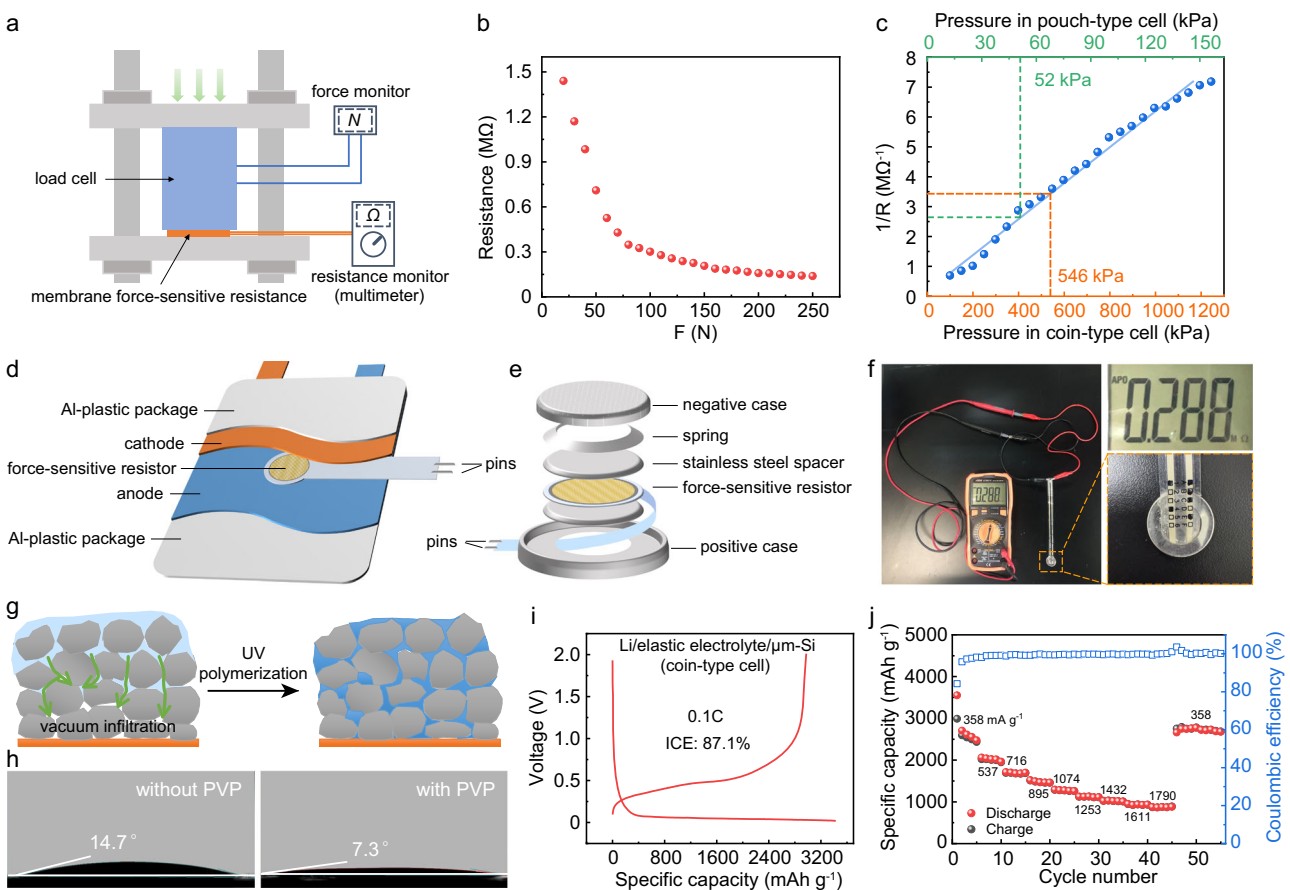

**Fig. 4 | Built-in pressure monitoring in cells and the fabrication of µm-Si anode with elastic electrolyte. a** Schematic of calibration of the membrane force-sensitive resistance (MFSR). The resistance of the MFSR could be read directly on the multimeter, and the applied force on the MFSR was monitored through a load cell. **b** The measured resistance to the applied force of the MFSR. **c** The fitted standard response curve (blue) of resistance to pressure of the MFSR. The built-in pressure of the coin-type and pouch-type cell is indicated with orange and green dashed line, respectively. Schematic of the home-made pressure sensing unit for (**d**) the pouch-type cell and (**e**) the coin-type cell. **f** Digital images of the built-in pressure measuring of the coin-type cell using the MFSR. Enlarged views of the

reading on the multimeter and the home-made pressure sensing unit are displayed on the right. **g** Schematic of the vacuum infiltration process of the precursor solution into the µm-Si electrode and the following UV polymerization process. The precursor solution penetrated and filled the porous electrode. After UV polymerization, the active materials in the porous electrode were surrounded and fettered by the elastic electrolyte. **h** The contact angle of the precursor solution on the µm-Si electrode without (left) and with (right) the surfactant PVP. **i** The initial discharge-charge curve and (**j**) rate performances of the solid-state µm-Si electrode in the coin-type Li/elastic electrolyte/µm-Si cell without external pressure. Source data are provided as a Source Data file.

anode material due to its high theoretical specific capacity (over 3700 mAh g$^{-1}$), low working potential (0.4 V versus Li$^+$/Li) and abundant reserves. Whereas, the immense volume change of Si during cycle impedes its practical application. Structure manipulation such as nano-crystallization can enhance the cycle stability of the Si electrode[13,34,35]. However, the sophisticated structural design greatly raises the cost. In comparison, µm-Si is considered as a more competitive candidate for the large-scale energy storage due to its low cost. Whereas, the µm-Si electrode suffers from rapid capacity decay in the cells with liquid electrolytes; and in the solid-state batteries, indispensable high stack pressure obstructs its practical application, as mentioned above. As a representative active material with promising prospects but tremendous volume change, µm-Si was selected as the experimental subject to evaluate the effectivity of using elastic electrolytes to enhance the cycle stability of solid-state batteries without external pressure.

To be more specific, the precursor solution with polymer monomers of the elastic electrolyte was dripped onto the Si electrode, followed by a vacuum infiltration process and then UV polymerization (Fig. 4g). In this way, the solid elastic electrolyte would enter into the µm-Si electrode and serve as both Li-ion transport channels and stress-buffering medium. As shown in Supplementary Fig. 17, the pristine electrode was porous, consisting of irregular Si particles in several micrometers. After the infiltration and polymerization of the electrolyte precursor solution, the elastic electrolyte wrapped around the Si granules, establishing unobstructed and efficient Li-ion transport network. To enhance the wettability of the precursor solution and facilitate its penetration into porous electrodes, trace amounts of surfactants polyvinylpyrrolidone (PVP) were introduced into the µm-Si electrode during the electrode preparation process. The detailed experimental approach can be found in the Materials and Methods. The contact angle of the precursor solution on the µm-Si electrode was reduced from 14.7° to 7.3° after the addition of PVP, implying easier wetting and infusing of the electrolyte (Fig. 4h). The fabricated Li/ elastic electrolyte/µm-Si coin-type cell delivered a high initial discharge capacity of 3413.6 mAh g$^{-1}$ at 0.1C (1C = 3579 mA g$^{-1}$), which approached the theoretical capacity of Si, with the coulombic efficiency of 87.1% (Fig. 4i). Moreover, the cell showed outstanding rate performance (Fig. 4j). The specific capacity achieved 890.5 mAh g$^{-1}$ even at a high current of 1880 mA g$^{-1}$. When the current returned to 376 mA g$^{-1}$, the specific capacity recovered to 2747.5 mAh g$^{-1}$. The high specific capacity and excellent rate behavior of the cell were attributed to the good contact between Si and the electrolyte. Even after cycles without external compressing, no observable detachment of the electrolyte from Si particles was detected, despite noticeable expansion of the µm-Si electrode (Supplementary Fig. 18).

## Electrochemical performances of the µm-Si electrode involving the elastic electrolyte

Galvanostatic cycle tests were carried out to assess the long-term stability of the solid-state µm-Si electrode free from external pressure. As displayed in Fig. 5a, the cell with the elastic electrolyte had an initial discharge capacity of 2081.6 mAh g$^{-1}$ at 0.2 C and retained a reversible specific capacity of 1039.7 mAh g$^{-1}$ at 0.4 C. The capacity retention reached 90.8% after 300 cycles with an average coulombic efficiency of 99.3%. Furthermore, as displayed in Supplementary Fig. 19, the Li/elastic electrolyte/µm-Si cell with a higher loading of 1.3 mg cm$^{-2}$ under no external stack pressure delivered an initial discharge capacity of 1377.8 mAh g$^{-1}$ with the coulombic efficiency of 83% at 0.2C (i.e. 0.9 mA cm$^{-2}$), and maintained a reversible specific capacity of 757 mAh g$^{-1}$ at elevated current density of 0.3 C (i.e. 1.4 mA cm$^{-2}$). In sharp contrast, the coin-type Li-In/Li$_6$PS$_5$Cl/µm-Si battery without additional pressure could not work at the first cycle (Supplementary Fig. 20a and Fig. 5a), and it suffered from a rapid capacity decay from 940.3 mAh g$^{-1}$ to 134 mAh g$^{-1}$ within 5 cycles even under a high stack pressure of 2 MPa (Supplementary Fig. 20b). These unsatisfactory

performances were similarly reported in the previous work, indicating that battery failure is common in solid-state cells involving inorganic electrolytes without high stack pressure[36]. The excellent cycle stability of the µm-Si with the elastic electrolyte should be attributed to the good elasticity and energy dissipation property of the electrolyte, which helped to maintain tight interface contact during cycle and impeded the disintegration of the µm-Si electrode. This perspective was further verified by the Finite Element Simulations (FES) on the stress evolution in the µm-Si anode. As illustrated in Fig. 5b (I), it was difficult for the rigid SSEs to dissipate stress caused by the expansion of Si, so that significant stress concentration could be observed during lithiation. By contrast, the elastic electrolyte could consume massive energy through the break of noncovalent bond, as analyzed above, leading to lower and more evenly distributed stress inside the µm-Si anode [Fig. 5b (II)]. In order to visually observe the structure change of the µm-Si electrode, non-destructive X-ray computed tomography (XR-CT) was carried out for the µm-Si/Li cells at the pristine state and after cycles. The initial µm-Si electrode with the elastic electrolyte possessed compact structure, which further confirmed the infiltration of the elastic electrolyte into the Si electrode (Supplementary Fig. 21). As shown in Fig. 5d(I)−(II), the Si granules remained in intimate contact with the electrolyte after cycle in the coin-type cell. Noted that the bright stripes were the artifacts, which were common in the XR-CT tests[37,38], rather than the real morphology of the µm-Si electrode. The integrity of the electrode was maintained ideally after repeated expansion and contraction of Si due to the outstanding deformation restore ability of the elastic electrolyte. Whereas, the violent volume change of Si brought about severe degradation of the µm-Si electrode with Li$_6$PS$_5$Cl electrolyte after cycle. Even though it was cycled under a high stack pressure of 10 MPa, large cracks and holes (marked in red) could be distinctly observed [Fig. 5d(III)−(IV)]. The gaps between Si particles and the electrolyte hindered the transportation of Li-ion and facilitated the disintegration of the electrode, and finally caused the failure of the cell.

Further assessment on the µm-Si electrode with the elastic electrolyte was conducted on full cells. Firstly, galvanostatic charge and discharge tests were carried out on the Li||LFP and Li|| LiNi$_{0.8}$Mn$_{0.1}$Co$_{0.1}$O$_2$ (NMC811) cells to assess the compatibility of the elastic electrolyte with cathodes. As illustrated in Fig. 5c, an initial discharge capacity of 147.4 mAh g$^{-1}$ was achieved by the Li/elastic electrolyte/LFP cell at 1 C (1C = 175 mA g$^{-1}$) with the coulombic efficiency of 97.1%, and it exhibited excellent cycle stability for over 400 cycles with a high-capacity retention of 97.3%. Furthermore, the cell displayed great rate capability (Supplementary Fig. 22). The reversible capacity reached 101.6 mAh g$^{-1}$ at a high current density of 875 mA g$^{-1}$ (5 C). The elastic electrolyte also showed compatibility toward NMC811 cathode. The Li/elastic electrolyte/NMC811 cell with 1 wt% LiPO$_2$F$_2$ as the additive in the cathode working under no external stack pressure delivered a specific discharge capacity of 207.1 mAh g$^{-1}$ with an initial coulombic efficiency of 85.4% and maintained a capacity retention of 77.4% after 90 cycles. (Supplementary Fig. 23). Coupling the well-performed µm-Si anodes and LFP cathodes, the µm-Si/elastic electrolyte/LFP full cells delivered a reversible specific capacity of 162.3 mAh g$^{-1}$ at 0.1C (calculated based on the mass of LFP) and maintained 151.4 mAh g$^{-1}$ at 0.5 C. The full cell operated stably for 100 cycles with negligible capacity decay (98.3% capacity retention, Fig. 5f). Moreover, it exhibited the longest lifespan (145 cycles with 80% capacity retention) among solid-state batteries relying solely on the built-in pressure of the coin-type cell (Fig. 5g), illustrating the effectiveness of the mechanical optimization strategy. Noted that this comparison was restricted to solid-state batteries involving high specific capacity anodes (including Li and Si) with inorganic and polymer SSEs, considering the more promising prospects of these battery systems.

Furthermore, the feasibility and safety of the µm-Si/elastic electrolyte/LFP full cell were tested and verified in the pouch cell configuration, of which the built-in pressure was 52 kPa originating from

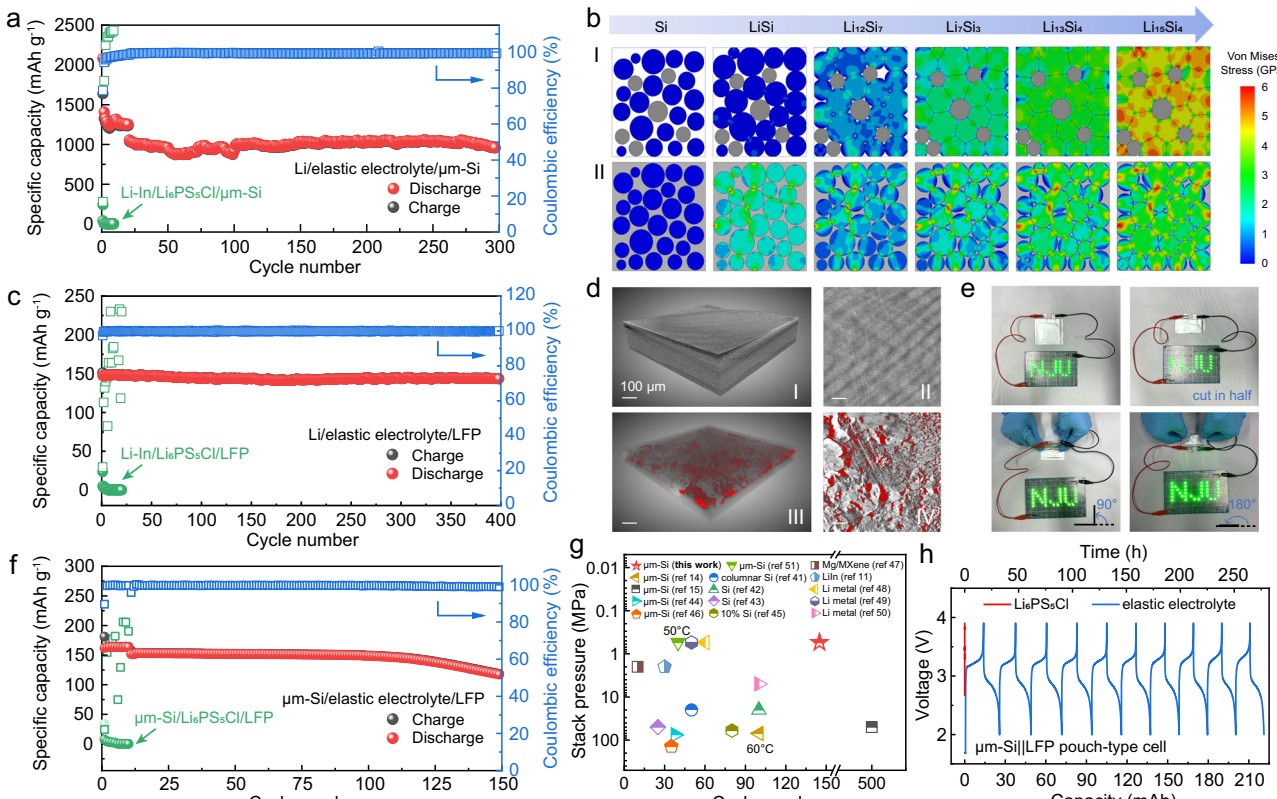

**Fig. 5 | Electrochemical performances of the µm-Si electrodes with the elastic electrolyte. a** The long-term cycling stability test of the solid-state µm-Si electrodes with the elastic electrolyte and $Li_6PS_5Cl$ electrolyte (green) in the coin-type Li|| µm-Si (Li-In||µm-Si for $Li_6PS_5Cl$ electrolyte) cell at 0.2 C and 0.4 C. **b** Stress distribution and evolution in the µm-Si electrode with (I) the $Li_6PS_5Cl$ solid-state electrolyte and (II) the elastic electrolyte simulated through the Finite Element Method. **c** The cycling stability test of the LFP cathode with the elastic electrolyte and $Li_6PS_5Cl$ electrolyte (green) in the coin-type Li||LFP cell. **d** Reconstructed 3D structure of the µm-Si electrode with (I)-(II) the elastic electrolyte and (III)-(IV) $Li_6PS_5Cl$ electrolyte after 20 cycles using XR-CT. The pores inside the electrode were represented in red. The scale bar was 100 µm. **e** Digital images of the µm-

Si/elastic electrolyte/LFP pouch cell lighting a series of green LED. Its ability to power the LED lights remained well even after being cut in half or bended to 90° and 180°. **f** The charge-discharge specific capacity and the corresponding coulombic efficiency of the coin-type µm-Si||LFP cells with the elastic electrolyte and $Li_6PS_5Cl$ electrolyte (green). **g** Comparison of cycle number and stack pressure of the µm-Si/elastic electrolyte/LFP cell in this work with other reported solid-state batteries[11,14,15,41–51] Noted that the cycle number of each cell is the number of the cycle with 80% capacity retention. **h** The charge-discharge curve of the µm-Si||LFP pouch cell with the elastic electrolyte (blue) and $Li_6PS_5Cl$ electrolyte (red) at 0.1 C. All the coin-type and pouch-type cells were tested without external pressure. Source data are provided as a Source Data file.

packing, as measured in Supplementary Fig. 14 and Fig. 4c. The homemade pouch cell with a discharge capacity of 11.7 mAh (corresponding to a high specific capacity of 140 mAh g⁻¹ calculated based on the mass of LFP) can operate charge and discharge cycles normally (Fig. 5h). By contrast, the µm-Si/$Li_6PS_5Cl$/LFP pouch cell barely possessed capacity. Furthermore, as the photos showing in Fig. 5e, a series of green light emitting diodes (LED) were successfully lighted by the µm-Si/elastic electrolyte/LFP pouch cell. Cutting and bending were exerted on the pouch cell and these abuse conditions did not jeopardize its ability to power the LED lights. The superb electrochemical performances and satisfactory safety of the µm-Si/elastic electrolyte/LFP cells verified the feasibility of the mechanical optimization strategy involving elastic electrolyte for the µm-Si electrode.

To sum up, the need for high stack pressure has long been a barrier to the practical implementation of solid-state batteries. Failures in solid-state batteries often result from poor contact between active materials and the solid electrolyte due to the volume changes that occur during cycling. To address this issue, this study introduces a mechanical optimization strategy involving an elastic electrolyte, enabling solid-state electrodes to function without the need for additional pressurization.

The elastic electrolyte comprises a highly solvated soft phase (poly-DMAM) and a poorly solvated rigid phase (poly-AM) in DEM (NMA-LiFSI), with an optimized molar ratio of 0.8 for the rigid phase.

The synergistic effect of these soft and rigid phases imparts the elastic electrolyte with exceptional mechanical properties, including outstanding stretchability (break elongation of 1160%), high fracture strength (1.7 MPa), shape memory capability, and remarkable energy dissipation features. Additionally, the ionic conductivity of the elastic electrolyte reaches $2 \times 10^{-3}$ S cm⁻¹ at room temperature. The electrolyte exhibits excellent electrochemical stability, as evidenced by its compatibility with Li metal and µm-Si anodes, along with a high oxidation potential of 4.5 V. Furthermore, the electrolyte is non-combustible and possesses self-healing property, ensuring the safety of solid-state batteries. Internal pressures within coin-type and pouch-type cells are monitored using embedded pressure sensors, with pressures recorded at 546 and 52 kPa, respectively. The elastic electrolyte is introduced into porous electrodes through vacuum infiltration and UV polymerization, with a trace amount of PVP acting as a surfactant to aid precursor solution penetration. As a result, solid-state µm-Si electrodes with the elastic electrolyte, tested in coin-type cells without additional pressurization, demonstrate outstanding long-term cycle stability, with 300 cycles achieved. Through XR-CT observation, it is evident that µm-Si granules maintained close contact with the elastic electrolyte without the formation of voids even after cycling. Similarly, the LFP cathode with the elastic electrolyte also performs stably for 400 cycles, retaining 97.3% of its initial capacity. Furthermore, the assembled µm-Si/elastic electrolyte/LFP full cell operates successfully

for 100 cycles, maintaining 98.3% capacity retention in the absence of external pressure. Even in a further reduced build-in pressure, the μm-Si/elastic electrolyte/LFP pouch cell delivers a discharge capacity of 11.7 mAh and powers a series of LEDs after being subjected to cutting and bending, showcasing excellent reliability and safety. In conclusion, by employing this technology with the solid elastic electrolyte, stable operation of solid-state batteries can be achieved without the need for external pressure, relying solely on the built-in pressure of coin cell or pouch-type cell. This strategy can address the stress concentration problem and related mechanical failure issues in solid-state cells, which are common due to the volume changes experienced by most active materials during battery cycling. This significantly advances the practical applications of solid-state batteries.

## Methods

### Synthesis of the elastic electrolyte
NMA (Aladdin) and LiFSI (DoDoChem) were mixed in different molar ratios and stirred until forming homogeneous mixtures in order to obtain DEMs. Considering the highest ionic conductivity of DEM with NMA:LiFSI = 4:1, this proportion was used in the subsequent synthesis of the elastic electrolytes. The elastic electrolytes were obtained by the UV polymerization of AM and DMAM in the DEM. Typically, 2.4 mol L$^{-1}$ AM (Aladdin), 0.6 mol L$^{-1}$ DMAM (Aladdin), 3 mmol L$^{-1}$ crosslinker $N,N$'-methylenebis(acrylamide) (Aladdin) and 1.5 mmol L$^{-1}$ photoinitiator 2-hydroxy−4'-(2-hydroxyethoxy)-2-methylpropiophenone (Aladdin) were added into the DEM and stirred to form homogeneous precursor solution. Then the precursor solution was poured on the fluorinated ethylene propylene (FEP) film and curing under UV light for 5 min to obtain the solid-state elastic electrolyte. The thickness of the electrolyte film could be controlled using the doctor blade. The elastic electrolytes with different ratios of rigid and soft phase were prepared with the same method, except that the proportions of AM and DMAM varied accordingly. All of the preparation process was completed in an Ar-filled glove box.

### Materials characterization
SEM characterization was conducted on a Hitachi SU8010 scanning electron microscope. The surfaces of the electrolyte were sprayed with gold before SEM testing due to low electronic conductivity of the sample. Raman tests were carried out on NTEGRA Spectra AFM Raman Confocal SNOM instrument with laser wavelength of 633 nm. SAXS was carried out on Xeuss 2.0 instrument with Cu X-ray radiator. The domain size of the microphase separation structure was calculated according to the equation

$$d = \frac{2\pi}{q} \quad (2)$$

where $q$ was the peak of the SAXS profile. IR spectra was collected using NEXUS870 (Thermo Fisher). AFM characterization was carried out on Asylum Research Cypher AFM instrument. DSC tests were conducted on DSC200F3 (Netzsch) from −100 °C to 0 °C under N$_2$ atmosphere. XR-CT characterizations were carried out on X-ray microscope (Xradia 620 Versa, Zeiss). The μm-Si electrode with the elastic electrolyte was cycled in the coin-type μm-Si/Li cell without additional pressurization (wherein the inner pressure was 546 kPa). The μm-Si electrode with Li$_6$PS$_5$Cl was cycled under a stack pressure of 10 MPa because it could not cycle normally under lower stack pressure. The battery samples were then removed out from the battery cases and sealed in Al-plastic film in the Ar-filled glove box for XR-CT characterizations.

### Mechanical property measurements
The tensile and compressive stress-strain experiments were conducted on TianYuan (TY-8000-A) Electronic Universal Test Instrument. To prepare samples for the tensile tests, the precursor solution was poured into a dumbbell-shaped Teflon mold with an effective size of 16 × 4 × 2 mm, followed by UV polymerization for 5 min. Samples for the compressive tests were cylinders (diameter = 1 cm, height = 1 cm) prepared through the same method. The tensile deformation rate was set as 100 mm min$^{-1}$ and the compressive deformation rate was set as 1 mm min$^{-1}$. The cyclic tests were performed at the same deformation rate with a rest time of 5 min between each cycle for the reconstruction of noncovalent bonds. All of the mechanical property measurements were performed in a dry room with a dew point of -40 °C.

### Ion transport property measurements
The ionic conductivity of DEMs and the elastic electrolyte film was calculated according to the equation

$$\sigma = \frac{L}{RS} \quad (3)$$

where $L$ was the thickness of the separator (for DEMs) or the elastic electrolyte membrane, $R$ was the bulk resistance, $S$ was the contact area of the electrolyte membrane (or the separator) and the blocking electrode. The EIS tests were conducted on Solartron1260/1287 with a frequency range from 1 MHz to 0.1 Hz and an amplitude of 10 mV.

The Li$^+$ transference number was measured through chronoamperometry method and EIS conducted on the Li/elastic electrolyte/Li symmetric cell and was calculated according to the equation

$$t_{Li^+} = \frac{I_s(\Delta V - I_0 R_0)}{I_0(\Delta V - I_s R_s)} \quad (4)$$

where $\Delta V$ was the applied potential, $I_0$ was the initial current, $I_s$ was the steady-state current, $R_0$ and $R_s$ were the resistances before and after the polarization, respectively.

### Cell assembly and electrochemical tests
Firstly, the μm-Si (Canrd New Energy Technology Co., Ltd.), carbon nanotubes (CNTs) and Super P with a weight ratio of 70:7.5:7.5 were ball-milled at 500 rpm for 10 h in Ar atmosphere to obtain the Si-C composite. Polyacrylic acid (PAA, molecular weight 450000, Aladdin) was dissolved in deionized water with a weight ratio of 5% to form the binder solution. PVP (molecular weight 58,000, Aladdin) served as the surfactant was added into the PAA aqueous solution in this step with a weight ratio of PVP: PAA = 15:1. Then the Si-C composite was added into the binder solution with the weight ratio of Si-C composite: PAA = 85:15 and formed homogenous slurry after magnetic stirring. After that, the slurry was coated on the Cu foil by doctor blade with the Si loading of 0.5–0.7 mg cm$^{-2}$. To fabricate the Li/elastic electrolyte/μm-Si cell, 50 μL precursor solution of the elastic electrolyte was dropped on the μm-Si electrode. After the full infiltration of the precursor solution, the electrode was exposed to UV light for 5 min for the polymerization of the electrolyte. Upon UV exposure, the photoinitiator generated the primary radicals via α-cleavage and triggered the opening of the C=C bonds in the monomers to form C−C· radicals, thereupon then a chain reaction is initiated until the completion of polymerization. The elastic electrolyte membrane was prepared as discussed above. Fluoroethylene carbonate in 5 wt% was added in the electrolyte to assist the formation of stable SEI. The Li foil (China Energy Lithium Co., Ltd.) was cut into disks with a diameter of 12 mm and used as the metal electrode without additional treatment. 2032 coin-type Li/elastic electrolyte/μm-Si cells were assembled by attaching the μm-Si electrode on one side of the electrolyte membrane, and attaching Li foil on the other side. As for the coin-type μm-Si/elastic electrolyte/LFP cells, LFP cathode was prepared by mixing LFP, acetylene black and PVDF in a weight ratio of 8:1:1 in $N$-methylpyrrolidone. The homogenous slurry was coated on the Al foil by doctor blade with the LFP loading of around 6 mg cm$^{-2}$. Then 50 μL precursor solution of the elastic electrolyte was dropped

on the LFP electrode, followed by the vacuum infiltration process and then UV polymerization. The cell assembling process was the same as that for Li/elastic electrolyte/μm-Si cells, except that the Li foil was changed to the LFP cathode. A similar preparation method was applied to fabricate the NMC811 cathode. To be more specific, the precursor solution of the elastic electrolyte was dripped onto the NMC811 cathode coated on the Al foil, followed by a vacuum infiltration process and then UV polymerization. For the μm-Si/elastic electrolyte/LFP pouch cells, the LFP cathode (40 mm × 40 mm), the elastic electrolyte membrane (45 mm × 45 mm) and the μm-Si anode (40 mm × 40 mm) were vacuum sealed in an aluminum-plastic package. The preparation and assembling processes of coin-type cells were completed in Ar-filled glove box and the assembling of pouch cells was carried out in the dry room with a dew point of −40 °C.

As to the Li-In /$Li_6PS_5Cl$/μm-Si and μm-Si/$Li_6PS_5Cl$/LFP cells, the μm-Si composite was prepared through ball-milling of μm-Si, $Li_6PS_5Cl$ and CNTs with a weight ratio of 6:3:1. The LFP composite cathode was obtained by mixing LFP, $Li_3InCl_6$ and vapor grown carbon fibers with a weight ratio of 6:3:1. $Li_3InCl_6$ was used in the LFP composite cathode in consideration of its good compatibility to LFP cathode[39]. During the fabrication of the Li-In /$Li_6PS_5Cl$/μm-Si battery, 100 mg $Li_6PS_5Cl$ was added into a poly(etherether-ketone) (PEEK) cylinder with an internal diameter of 10 mm and then mechanically pressed under 300 MPa for 5 min. After that, 1 mg μm-Si composite was distributed evenly on one side of the $Li_6PS_5Cl$ pallet and followed by a mechanical pressing under 300 MPa for another 5 min. Then Li-In was attached to the other side of the $Li_6PS_5Cl$ pallet and the fabricated cell was compressed under 150 MPa for 6 h. The μm-Si/$Li_6PS_5Cl$/LFP cells were assembled with the similar method. First, 50 mg $Li_3InCl_6$ and 50 mg $Li_6PS_5Cl$ was pressed under 300 MPa for 5 min in the PEEK cylinder. Then the LFP composite cathode was spread on the $Li_3InCl_6$ side of the electrolyte pallet and the μm-Si composite anode was spread on the $Li_6PS_5Cl$ side. After that, the μm-Si/$Li_6PS_5Cl$/LFP cell was compressed under 150 MPa for 6 h. Finally, the cells were transferred from the cell molds to the coin-type cells (or the required external pressure was applied on the cell molds) for the electrochemical tests.

The galvanostatic discharge and charge tests were carried on NEWARE Battery Test System (CT-4008T, Shenzhen, China). The cut-off voltages for the Li‖μm-Si cells were 0.05 V (0.02 V for the first cycle) and 2 V during lithiation and delithiation, respectively. Note that the μm-Si electrodes for the μm-Si‖LFP full cells were lithiated and delithiated for one time in the Li‖μm-Si cells to compensate for the irreversible loss of Li during the first cycle. After that, the cycled μm-Si electrodes were used to assemble the μm-Si‖LFP cells without additional treatment. The μm-Si‖LFP full cells were cycled within the voltage range of 2–3.9 V. No external stack pressure was applied to the cells during the electrochemical tests unless otherwise explicitly specified. All of the electrochemical tests were carried out at room temperature (25°C) unless otherwise explicitly specified.

**Finite element simulations**

The stress distribution and evolution in the μm-Si electrode with $Li_6PS_5Cl$ and the elastic electrolyte were simulated through the finite element method. Firstly, models with randomly distributed Si spheres were established to simulate the expansion of Si during lithiation process. The models were constructed based on the linear elasticity assumption, of which the relevant parameters of Si during lithiation can be found in the previous work[40]. The equation for the calculation satisfied Newton's second law and followed

$$\nabla \cdot \boldsymbol{\sigma} + \mathbf{f} = \rho \frac{\partial^2 \mathbf{u}}{\partial t^2} \tag{5}$$

where $\sigma$ was the stress tensor, $\mathbf{f}$ was the body force, $\rho$ was the density and $\mathbf{u}$ was the displacement vector. The implicit algorithm method was used and gradually iterated to the maximum expansion of Si to avoid convergence difficulties.

## Data availability

The datasets generated and analysed in this work are included in this article and Supplementary Information. Source data are provided with this paper.

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

## Acknowledgements

This research was supported by the National Key R&D Program of China (2021YFB3800300-P.H. and H.Z.), the National Natural Science Foundation of China (22179059-P.H., 22239002-H.Z.), Key R&D project funded by department of science and technology of Jiangsu Province (BE2020003-P.H.), science and technology innovation fund for emission peak and carbon neutrality of Jiangsu province (BK20231512-P.H., BK20220034-H.Z.), Postgraduate Research & Practice Innovation Program of Jiangsu Province (KYCX23_0145-H.P.).

## Author contributions

P.H. conceived the idea and supervised the research. H.P., L.W., and Y.S. conducted the experiments. C.S. carried out the scanning electron microscopy characterization. H.P. and P.H. analyzed the experiment results and wrote the manuscript. H.P., P.H., S.Y., and H.Z. discussed the results and commented on the manuscript.

## Competing interests

The authors declare no competing interests.
