## [Peer Review File · Nature Communications]

A solid-state lithium-ion battery with micron-sized silicon anode operating free from external pressureREVIEWER COMMENTS

Reviewer #1 (Remarks to the Author):

The authors present the design of an elastic polymer mixed with N-methylacetamide-LIFSI solution as an electrolyte for Si|LFP batteries and demonstrate its operation under zero external pressure. Reducing external pressure is crucial for high energy density batteries, and this work shows decent electrochemical performance in the specific system. However, I have two major concerns.

Firstly, the electrolyte seems to resemble a gel polymer electrolyte rather than a solid electrolyte since a liquid solution (N-methylacetamide-LIFSI) is used. With the presence of the liquid phase, high pressure is not necessary to maintain good ionic conduction. Poly(dimethylacrylamide)-based electrolytes have been extensively studied in the battery field (Macromolecules 1996, 29, 1, 143–155; Solid State Ionics 2003, 157, 233-239; Electrochimica Acta 1995, 40, 2417-2420).

Secondly, for high-energy silicon batteries, it is recommended to pair NMC with a Si-based anode. LFP is not an energy-dense option and lacks general interest (Science 2021, 373, 1494–1499).

Fig. 1: The function of the N-methylacetamide-LIFSI mixture is missing.

Fig. 2b: Is N-methylacetamide chemically bonded/crosslinked on the polymer backbone? Please provide experimental evidence if so.

Fig. 2d: How does the stretching recovery capability help with the exemption of external pressure?

UV curing was used for the in-situ preparation of the silicon anode. The light was supposed to be blocked by the Si electrode. The bottom part of the electrode is not exposed to the light. How would the UV curing be processed? It is unclear if there are liquid electrolytes at the bottom of the Si electrode. Additionally, the Si loading of 0.5-0.7 mg cm⁻² is low for practical applications.

"Fluoroethylene carbonate in 5wt% was added to the electrolyte to assist the formation of a stable SEI." This statement confuses me. It appears to be a liquid additive that was not mentioned in the main text. Furthermore, there seems to be liquid SEI formation in the system.

On the cathode side, there is no polymer electrolyte present. How are Li ions conducted without an electrolyte? I believe liquid may diffuse into the cathode.

Figure 5: The control samples of LiPSCI are not reasonable, as they consist of pure solid-state electrolyte, while the reported polymers contain a liquid phase.

Reviewer #2 (Remarks to the Author):

This work provides a solution to address the issue with the internal void formation in Si anode. And poor contact due to volume change by using an elastic solid electrolyte. A stable cycle performance was reported in a μm -Si anode without additional pressure. The mechanical design, which leverages microphase separation of soft and rigid phases, enables the enhanced stretchability, fracture strength, shape memory capacity, and energy dissipation properties. This manuscript is well organized and systematically studied, the innovation is good. There are some questions for authors to clarify before it can be accepted.

1. It has been previously reported that combining a silicon anode with high elastic and self-healing polymer electrolytes enables stable cycling, despite the significant volume expansion and shrinkage of silicon particle (e.g., Q. Huang et al., Nat. Commun., 10, 5586 (2019); C. Wang et al., Nat. Chem., 5, 1042-1048 (2013)). It is advisable to reference these prior studies while also highlighting the originality of your own work in the Introduction section.

2. The microphase separation in copolymer to increase mechanical strength is typically achieved using block copolymer. Is there a specific reason why a random copolymer was used in this work? Additionally, what are the monomer reactivity ratios of AM and DMAM monomer in a random copolymerization?

3. The polymer electrolyte with high ionic conductivity of $2 \times 10^{-3} \text{ S cm}^{-1}$ and reinforced mechanical properties is expected to demonstrate improved Li plating and stripping performance at elevated current density, as well as superior Li deposition, in comparison to the results shown in Fig. 3 (0.1 mA cm^{-2} , 0.1 mAh cm^{-2}). The rate performance of should be provided to further highlight its properties.

4. In Fig. 3b, it seems that the interfacial resistance is larger compared to the bulk resistance in Li symmetrical cell. What could be the reason for this?

5. In Fig. 5a, the cycle performance of $\mu\text{m-Si}$ half-cell was evaluated at 0.4 C with a lower cutoff voltage of 50 mV after the initial cycles, demonstrating a discharge capacity of 1039 mAh g^{-1} . However, a higher capacity of $\sim 2000 \text{ mAh g}^{-1}$ can be expected with a lower cutoff voltage of 50mV and low overpotential. How would the cycle performance be affected by a reduced C-rate and a higher discharge capacity, considering more severe expansion and shrinkage of Si particle?

6. The computational details of the finite element method to generate Figure 5b is not provided. Please add.

7. What is the capacity ratio of LFP to $\mu\text{m-Si}$ in an LFP/ $\mu\text{m-Si}$ full cell? This information is crucial as the utilization rate of Si anodes is considered to have a significant impact on the cycle characteristics of full cells.

Reviewer #3 (Remarks to the Author):

The need for external pressure is a major obstacle to the application of all-solid-state batteries in practice, but the authors raised an effective strategy to alleviate this issue, with convincing demonstration using the micrometer-Si anode. The reviewer considers it as a highly important contribution to the all-solid-state battery community. However, before it can be accepted for publication, the following issues need further clarification.

1. The authors indicate that the synthesis of the elastic polymer electrolyte is conducted in Ar-filled glovebox. Is it unstable in ambient air? Which component in air will react with this material? Is it possible to improve its air stability in future studies? This directly influences the production cost, which is also rather important to the successful commercialization of all-solid-state batteries.

2. The authors claim that their elastic polymer electrolyte shows an oxidation potential of 4.5 V vs. Li/Li⁺, but the cell for cycling tests utilizes a 3 V-class material, LiFePO₄, as the cathode. The reviewer would like to see the cycling performance of the cell where the elastic polymer electrolyte is paired with the 4 V-class cathodes like LiCoO₂ or LiNi_{0.8}Mn_{0.1}Co_{0.1}O₂.

3. The authors specified the areal mass loading of micrometer-Si in their cells, but the areal capacity (in mAh cm⁻²) that can be actually achieved in the cell is more meaningful. The desired mechanical properties reported here could also be helpful in reaching higher areal capacity. Therefore, the reviewer suggests the authors adjust the mass loading of micrometer-Si and investigate the maximum areal capacity that the micrometer-Si anode can achieve in the μm-Si/elastic electrolyte/Li cell with decent cycling stability.

Responses to Reviewers' Comments

We would like to thank all reviewers for their constructive comments and suggestions, which has in our view significantly raised the quality of the manuscript (NCOMMS-23-50616-T). We have modified the manuscript accordingly, and listed the detailed corrections below point by point for each reviewer. All revised portions have been marked in yellow in the revised manuscript. The main corrections and the responses to the reviewers' comments are as follows.

Point-to-Point Responses

Reviewer #1:

The authors present the design of an elastic polymer mixed with N-methylacetamide-LIFSI solution as an electrolyte for Si/LFP batteries and demonstrate its operation under zero external pressure. Reducing external pressure is crucial for high energy density batteries, and this work shows decent electrochemical performance in the specific system. However, I have two major concerns.

Firstly, the electrolyte seems to resemble a gel polymer electrolyte rather than a solid electrolyte since a liquid solution (N-methylacetamide-LIFSI) is used. With the presence of the liquid phase, high pressure is not necessary to maintain good ionic conduction. Poly(dimethylacrylamide)-based electrolytes have been extensively studied in the battery field (Macromolecules 1996, 29, 1, 143–155; Solid State Ionics 2003, 157, 233-239; Electrochimica Acta 1995, 40, 2417-2420).

Secondly, for high-energy silicon batteries, it is recommended to pair NMC with a Si-based anode. LFP is not an energy-dense option and lacks general interest (Science 2021, 373, 1494–1499).

Response: We thank the reviewer for the approval of the importance of our work and the affirmative comment on electrochemical performances of the batteries in the manuscript.

Regarding to the first concern from the reviewer, please allow me to point out that the deep eutectic mixture (DEM) consisting of N-methylacetamide and LiFSI has great differences from conventional organic liquid electrolytes. The DEM forms as a result of the intermolecular interactions between the solid-state hydrogen bond donor (N-methylacetamide) and acceptor (LiFSI), which makes DEM have similar properties to molten salts including incombustibility and low vapor pressure^{1,2}. Therefore, the elastic electrolyte proposed in our work possesses higher safety than the normal gel polymer electrolytes containing liquid solvents and presents nonflammability (see the combustion tests in Fig. 2j). Moreover, the elastic electrolyte features much superior mechanical properties (excellent stretchability and compressibility, desirable fatigue resistance, good deformation resiliency and unique energy dissipation characteristic) than gel electrolytes, which is crucial to realize the stable operation of the batteries without external pressure, especially those with large volume change.

It is well-acknowledged that there are trade-offs between the ionic conductivity and mechanical properties of polymer electrolytes. Although the adoption of liquid phase in gel electrolytes is beneficial to the Li⁺ conduction, it usually comes at the cost of mechanical strength³. For this reason,

despite of the fine ionic conductivity of gel polymer electrolytes, few works have succeeded in constructing well-performed $\mu\text{m-Si}$ anode with gel electrolytes. There were some attempts using nanostructured Si pairing with gel electrolytes. For example, poly(ethylene glycol)diglycidylether crosslinked with diamino-poly(propylene oxide) immersing in 1 M LiPF_6 solution was investigated as the gel electrolyte for nano-Si anode⁴. Triacrylate-based gel polymer electrolytes combined with mesoporous Si enabled the Li-Si battery to function for 100 cycles⁵. In these works, sophisticated nanostructure designs were still necessary to alleviate the volume change of Si due to the poor mechanical properties of the gel polymer electrolytes. However, the volumetric capacity of nano-Si is much smaller than that of $\mu\text{m-Si}$ due to its lower tap density⁶. Besides, the tedious structural design greatly raises the cost.

For the second concern from the reviewer, it is undeniable that NMC is a promising cathode material considering its high theoretical specific capacity. Whereas, it is hard for NMC to deliver satisfying actual specific capacity in the long-term cycle in solid-state batteries. The specific capacity of the NMC811 is calculated to be $\sim 80 \text{ mAh g}^{-1}$ in the $\mu\text{m-Si/NMC}$ cell in the mentioned reference, which is lower than that of LFP in our work (151.4 mAh g^{-1} at 0.5C for the long-term cycle test).

We agree with the reviewer that NMC is a more energy-dense cathode material and may raise general interest. **We have applied the elastic electrolyte in the NMC cathode and conducted galvanostatic cycle tests on the NMC/elastic electrolyte/Li cells without external pressure.** To fabricate the NMC electrode with the elastic electrolyte, the precursor solution of the elastic electrolyte was dripped onto the NMC cathode coated on the Al foil, followed by a vacuum infiltration process and then UV polymerization. As illustrated in Fig. R1, the cell possessed a specific capacity of 208.6 mAh g^{-1} with an initial coulombic efficiency of 82.6% and functioned normally for dozens of cycles, but the cycle stability still needed further improvements. It can be seen from Fig. R2 that NMC particles retained in tight contact with the elastic electrolyte after cycles, which ensured the rapid Li^+ transport inside the cathode. Consequently, it is reasonable to infer that the capacity decay was not caused by contact failure between the electrolyte and active materials, even though the cells are cycled with the exemption of external pressure.

Fig. R1. (a) The charge-discharge curve of the 1st, 5th, and 10th cycle and (b) the cycling stability test of the NMC/elastic electrolyte/Li cell without external pressure.

Fig. R2. Scanning electron microscopy images of the NMC electrode with the elastic electrolyte after cycles.

The capacity decay may be caused by the fact that the charge cutoff voltage of the cell (4.4 V versus Li⁺/Li) was quite close to the upper limit of the electrochemical stability window of the elastic electrolyte (4.5 V versus Li⁺/Li). In order to further enhance the electrochemical performances of the NMC/elastic electrolyte/Li cell, we employed 1 wt% LiPO₂F₂ as the additive in the elastic electrolyte in the cathode to construct a more stable cathode electrolyte interphase (CEI)⁷. The modified NMC/elastic electrolyte/Li cell working under no external stack pressure delivered a specific discharge capacity of 207.1 mAh g⁻¹ with an increased initial coulombic efficiency of 85.4%. Moreover, the cycle stability was meliorated and the cell maintained a capacity retention of 77.4% after 90 cycles (Fig. R3). The improved electrochemical performances were attributed to formation of the more robust CEI. It is reasonable to speculated that optimizing the type and amount of the additive could further enhance the compatibility of the elastic electrolyte with NMC. Whereas, this work is meant to focus on using the elastic electrolyte to tackle the mechanical failure issue of batteries working without high stack pressure. Further optimization of the electrochemical compatibility of the elastic electrolyte with NMC cathode remains a future work.

Fig. R3. (a) The charge-discharge curve of the 1st, 5th, and 10th cycle and (b) the cycling stability test of the Li/elastic electrolyte/NMC cell with 1 wt% LiPO₂F₂ as the additive in the cathode tested without external stack pressure.

Fig. 1: The function of the N-methylacetamide-LIFSI mixture is missing.

Response: We thank the reviewer for pointing this out. The N-methylacetamide-LiFSI mixture (DEM) played a role in conducting Li⁺ and assisted with the formation of the phase-separate structure in the elastic electrolyte. Fig. 1 is a three-dimensional diagram describing the failure mechanism of solid-state batteries and the corresponding solutions. The function of Li⁺ conduction has been indicated by arrows. However, for the sake of conciseness, detailed depiction of the molecular interactions between the DEM and the copolymer network, which lead to the phase-separation structure, have not been included in Fig. 1., but in Fig. 2a. **According to the reviewer's**

comment, we have redrawn Fig. 2a and added a schematic diagram in Supplementary Fig. 2b to show N-methylacetamide molecules in the elastic electrolyte, as well as to clearly demonstrate the molecular structure and intermolecular interactions of the N-methylacetamide-LiFSI mixture; please see Fig. 2a in the revised manuscript and Supplementary Fig. 2b in the revised Supplementary Information. Moreover, we add a more explicit description about the function of the DEM in the formation of the phase-separate structure; please see page 5 in the revised manuscript. It has been highlighted in yellow in the revised manuscript and copied here: “The rigid and soft phase copolymerize randomly in the electrolyte and a bicontinuous phase-separate network forms due to the disparate miscibility of poly-DMAM and poly-AM with DEM.”

Fig. 2b: Is N-methylacetamide chemically bonded/crosslinked on the polymer backbone? Please provide experimental evidence if so.

Response: We thank the reviewer for this comment. According to this comment, we conduct Fourier transform infrared (FTIR) and Raman measurements on the precursor solution of the elastic electrolyte and the polymer after UV polymerization. From the results of the FTIR and Raman spectra, it can be seen that N-methylacetamide was not bonded or crosslinked on the polymer backbone. As shown in Fig. R4, the C=O stretching vibration peak of N-methylacetamide at 1652 cm^{-1} did not shift or weaken after polymerization. Furthermore, the Raman peaks corresponding to N-methylacetamide remained unchanged after polymerization (Fig. R5a). Instead, the Raman peak at 1625 cm^{-1} belonging to the C=C in the monomers vanished due to the opening of the C=C (Fig. R5b).

Fig. R4. FTIR spectroscopy of the N-methylacetamide-LiFSI mixture, the precursor solution of the elastic electrolyte and the polymer after UV polymerization.

Fig. R5. (a) Raman spectra of the N-methylacetamide, the N-methylacetamide-LiFSI mixture and the elastic

electrolyte after UV polymerization. (b) Raman spectra the precursor solution of the elastic electrolyte and the polymer after UV polymerization.

Fig. 2d: How does the stretching recovery capability help with the exemption of external pressure?

Response: We thank the reviewer for this comment. As depicted in Fig. 1 in the manuscript, the elastic electrolyte surrounds Si particles and fills the internal holes of the fabricated $\mu\text{m-Si}$ electrode. During the lithiation process, the elastic electrolyte filling in between Si particles is subjected to compression due to the volumetric expansion of Si, whereas the elastic electrolyte adhering to the surface of Si undergoes stretching. Conversely, Si granules shrink during delithiation and the deformed elastic electrolyte restores accordingly. Therefore, both the stretching and compressing recovery capability of the elastic electrolyte have an impact on the structural stability of the $\mu\text{m-Si}$ electrode working without extra stack pressure. By introducing an elastic electrolyte with outstanding stretching and compressing recovery capability, the intimate contact between the electrolyte and the active materials can be readily maintained despite of the volume fluctuation of the active materials. In this way, rapid Li^+ transport can be realized dispensing with external pressure.

UV curing was used for the in-situ preparation of the silicon anode. The light was supposed to be blocked by the Si electrode. The bottom part of the electrode is not exposed to the light. How would the UV curing be processed? It is unclear if there are liquid electrolytes at the bottom of the Si electrode. Additionally, the Si loading of 0.5-0.7 mg cm^{-2} is low for practical applications.

Response: We thank the reviewer for this comment. 2-hydroxy-4'-(2-hydroxyethoxy)-2-methylpropiophenone (Irgacure 2959) used in our work is a free radical photoinitiator that can generate the primary radicals via α -cleavage after UV exposure⁸. The primary free radicals can trigger the opening of $\text{C}=\text{C}$ bonds⁹ in the monomers to form $\text{C}-\text{C}\cdot$ radicals, thereupon then a chain reaction is initiated until the completion of polymerization. The UV light serves only as a triggering condition for the start of the chain reaction. Therefore, even though the bottom part of the electrode was not directly exposed to the light, the polymerization of monomers could be realized successfully through the free radical chain reaction. **In response to the reviewer's comment, we add a statement about the UV initiated chain polymerization in the "Cell assembly and Electrochemical tests" part in Methods; please see page 16 in the revised manuscript.** It has been highlighted in yellow in the revised manuscript and copied here: "Upon UV exposure, the photoinitiator generated the primary radicals via α -cleavage and triggered the opening of the $\text{C}=\text{C}$ bonds in the monomers to form $\text{C}-\text{C}\cdot$ radicals, thereupon then a chain reaction is initiated until the completion of polymerization."

We are in full agreement with the reviewer that active material loading is one of the key indicators for practical batteries. The Si loading of 0.5-0.7 mg cm^{-2} corresponds to theoretical areal capacity of 1.8-2.5 mAh cm^{-2} , and the actual capacity reached 2.1 mAh cm^{-2} (3413.6 mAh g^{-1} , Fig. 4i) in our work. Such capacity (or lower capacity) has been frequently-used in innovation researches for lithium-ion batteries¹⁰⁻¹⁴, considering that high loading is not the main emphasis of these studies. **Nevertheless, in response to the comment from the reviewer, we conduct galvanostatic discharge and charge tests on solid-state $\mu\text{m-Si}$ with higher loading of 1.3 mg cm^{-2} under no external stack pressure.** The $\mu\text{m-Si}$ /elastic electrolyte/Li cell was tested at 0.2C and 0.3C (1C=3579 mA g^{-1}). As illustrated in Fig. R6, the cell delivered an initial discharge capacity of 1377.8 mAh g^{-1} with the coulombic efficiency of 83% at 0.2C (i.e. 0.9 mA cm^{-2}). With an elevated current

density to 0.3C (i.e. 1.4 mA cm⁻²), the reversible specific capacity decreased to 757 mAh g⁻¹. The reduced specific capacity may be caused by the enlarged overpotential of the Li counter electrode. It can be seen from the rate performance of the Li/elastic electrolyte/Li symmetric cell that the overpotential increased with the current density (Fig. R7). After 50 cycles, the μm-Si/elastic electrolyte/Li cell maintained 631.3 mAh g⁻¹, corresponding to a capacity retention of 83.4%. Although the cycle life of the μm-Si/elastic electrolyte/Li cell is curtailed with the increase of mass loading, it can be seen that the proposed elastic electrolyte remains effective to enhance the stability of the high-loading solid-state μm-Si electrode operating without external pressure.

Fig. R6. The galvanostatic discharge and charge test on the μm-Si/elastic electrolyte/Li cell with a μm-Si loading of 1.3 mg cm⁻² at 0.2C and 0.3C without external stack pressure.

Fig. R7. Rate performance of the Li/elastic electrolyte/Li symmetric cell with an areal capacity of 0.1 mAh cm⁻² under no external stack pressure.

"Fluoroethylene carbonate in 5wt% was added to the electrolyte to assist the formation of a stable SEI." This statement confuses me. It appears to be a liquid additive that was not mentioned in the main text. Furthermore, there seems to be liquid SEI formation in the system.

Response: We thank the reviewer for this comment. Fluoroethylene carbonate has been widely employed as a sacrificial additive in the electrolyte to form a stable SEI on the Si anode¹⁵. The reductive decomposition of fluoroethylene carbonate occurs at a high potential of 1.3 V versus Li⁺/Li⁰¹⁶, which is higher than the lithiation potential of Si. That means the liquid additive fluoroethylene carbonate will decompose prior to the lithiation of the Si anode. The main decomposition products of fluoroethylene carbonate are solid-state species including inorganic LiF

and organic polycarbonates and CHF-OCO₂ compounds¹⁷⁻¹⁹. In our work, a very small amount of fluoroethylene carbonate was used to assist the formation of robust SEI on the Si anode and it decomposed completely during the first few cycles. That was why the coulombic efficiency of the $\mu\text{m-Si}$ /elastic electrolyte/Li cell was 87.1% at the first cycle and rose to higher than 99.5% within 10 cycles. Moreover, the addition of fluoroethylene carbonate itself does not necessarily lead to a long-term stability of the Si anode, as the reported Li-Si cells with 2-10 wt% fluoroethylene carbonate showed degradation within 50 cycles^{19, 20}. Therefore, it is reasonable to attribute the superior cycle stability of the Si anode (300 cycles with 90.8% capacity retention) to the outstanding mechanical properties of the elastic electrolyte.

On the cathode side, there is no polymer electrolyte present. How are Li ions conducted without an electrolyte? I believe liquid may diffuse into the cathode.

Response: We thank the reviewer for the careful reading of our manuscript. The elastic electrolyte was integrated into the cathode using the same method as in the Si anode. To be specific, 50 μL precursor solution of the elastic electrolyte was dropped on the $\mu\text{m-Si}$ electrode. After the full infiltration of the precursor solution, the electrode was exposed to UV light for 5 minutes for the polymerization of the electrolyte. **We have added a detailed description of the cathode fabrication in the “Cell assembly and Electrochemical tests” part in Methods; please see page 16-17 in the revised manuscript.** It has been highlighted in yellow in the revised manuscript and copied here: “*The homogenous slurry was coated on the Al foil by doctor blade with the LFP loading of around 6 mg cm⁻². Then 50 μL precursor solution of the elastic electrolyte was dropped on the LFP electrode, followed by the vacuum infiltration process and then UV polymerization.*”

Figure 5: The control samples of LiPSCl are not reasonable, as they consist of pure solid-state electrolyte, while the reported polymers contain a liquid phase.

Response: We thank the reviewer for this comment. The worry from the reviewer is completely understandable because the deep eutectic mixtures (DEM) of N-methylacetamide and LiFSI is liquid due to the strong intermolecular interactions, although both N-methylacetamide and LiFSI are solid-state at ambient temperature. However, the DEM has great differences from conventional organic liquid electrolytes. It forms as a result of the intermolecular interactions between the solid-state hydrogen bond donor (N-methylacetamide) and acceptor (LiFSI), which makes DEM have similar properties to molten salts including incombustibility and low vapor pressure^{1, 2}.

In addition, the DEM is not used independently as the electrolyte in our work. In the fabricated elastic electrolyte, the copolymer molecular chains crosslink through hydrogen bonds, while the DEM is constrained within the crosslinked polymer network, rendering it unable to flow like a free liquid. The elastic electrolyte appeared characteristics of solids including stretchability and compressibility, high fracture strength and deformation recovery capability. Even under abuse conditions (such as uniaxial tensile and compressive tests and the cutting tests), no leakage of liquid from the elastic electrolyte was detectable (see Fig 2d, 2g, 2i, Supplementary Fig. 10 and Supplementary Video). Therefore, although the elastic electrolyte contains DEM, it is rational to regard it possessing the characteristics of solid-state electrolytes. The selection of batteries with the LPSCl electrolyte as the control samples was out of the above consideration.

Reviewer #2:

This work provides a solution to address the issue with the internal void formation in Si anode. And poor contact due to volume change by using an elastic solid electrolyte. A stable cycle performance was reported in a μm -Si anode without additional pressure. The mechanical design, which leverages microphase separation of soft and rigid phases, enables the enhanced stretchability, fracture strength, shape memory capacity, and energy dissipation properties. This manuscript is well organized and systematically studied, the innovation is good. There are some questions for authors to clarify before it can be accepted.

Response: We sincerely appreciate the reviewer for this positive comment on the systematicness and originality of our work.

1. It has been previously reported that combining a silicon anode with high elastic and self-healing polymer electrolytes enables stable cycling, despite the significant volume expansion and shrinkage of silicon particle (e.g., Q. Huang et al., Nat. Commun., 10, 5586 (2019); C. Wang et al., Nat. Chem., 5, 1042-1048 (2013)). It is advisable to reference these prior studies while also highlighting the originality of your own work in the Introduction section.

Response: We thank the reviewer for the thoughtful suggestion. The references mentioned by the reviewer reported elastic and self-healing polymers as the protective layer on the Si anode working with organic liquid electrolytes. The electrochemical performances of these Si electrodes were greatly enhanced due to the circumvention of the structural collapse during cycling. Thus, the effectiveness of using elastic media to alleviate the Si electrode degradation was validated. Unfortunately, flammable organic liquid electrolytes were used in these previous works. In our work, nonflammable solid-state elastic electrolyte was designed and served as both the mechanical cushion and the electrolyte simultaneously. **In response to the reviewer's advice, the suggested relevant references have been added as the 17th and 18th reference and we have added the relevant discussion in the Introduction section; please see page 2-3 in the revised manuscript.** It has been highlighted in yellow in the revised manuscript and copied here: “Nevertheless, previous works have validated the effectiveness of using elastic media to alleviate the Si electrode degradation in the cells with liquid electrolytes^{17, 18}. Therefore, it is reasonable to speculate that even though the elastic solid electrolyte cannot completely prevent such deformations and fractures, it can effectively encase the active material when volume changes occur, thereby maintaining the efficient Li-ion transport.”

2. The microphase separation in copolymer to increase mechanical strength is typically achieved using block copolymer. Is there a specific reason why a random copolymer was used in this work? Additionally, what are the monomer reactivity ratios of AM and DMAM monomer in a random copolymerization?

Response: We thank the reviewer for this comment. As the reviewer points out, the microphase separation inside the elastic electrolyte is the key to its superior mechanical properties. The reason for the selection of a random copolymer is that different segments play distinct roles. To be more specific, the AM-rich segments contain abundant hydrogen bonds (as shown in Fig. 2a) which contribute to high strength and energy dissipation property; while the DMAM-rich segments feature weak interchain interactions and thus induce high deformability. **We have added the relevant**

discussion in the manuscript and it has been highlighted in yellow; please see page 7 of the revised manuscript.

In addition, the monomer reactivity ratios of AM and DMAM is approximately 0.78 and 1.11, respectively²¹. Both of the reactivity ratios are close to 1, therefore contributing to a random copolymerization behavior of the polymer network²².

3. The polymer electrolyte with high ionic conductivity of $2 \times 10^{-3} \text{ S cm}^{-1}$ and reinforced mechanical properties is expected to demonstrate improved Li plating and stripping performance at elevated current density, as well as superior Li deposition, in comparison to the results shown in Fig. 3 (0.1 mA cm^{-2} , 0.1 mAh cm^{-2}). The rate performance of should be provided to further highlight its properties.

Response: We thank the reviewer for the insightful comment. According to the reviewer's suggestion, we provide the rate performance of the Li/elastic electrolyte/Li symmetric cell without external stack pressure and the Li plating and stripping performance with a higher current density. As demonstrated in Fig. R8a, the overpotential of the symmetric cell increased with the elevated current density. Nonetheless, no indication of shortage arose even when the current density was up to 1 mA cm^{-2} . Furthermore, the symmetric cell realized a stable operation for over 350 hours at 0.2 mA cm^{-2} , 0.2 mAh cm^{-2} (Fig. R8b). We have added these data and the relevant discussion in the revised manuscript; please see page 8 and Supplementary Fig. 13.

Fig. R8. (a) Rate performance of the Li/elastic electrolyte/Li symmetric cell with an areal capacity of 0.1 mAh cm^{-2} . (b) Galvanostatic Li plating and stripping profiles of the Li/elastic electrolyte/Li symmetric cell at 0.2 mA cm^{-2} , 0.2 mAh cm^{-2} . The symmetric cells were tested without external stack pressure.

4. In Fig. 3b, it seems that the interfacial resistance is larger compared to the bulk resistance in Li symmetrical cell. What could be the reason for this?

Response: We thank the reviewer for this comment. In the Li symmetric cell, the bulk resistance is usually determined by the ionic conductivity of the electrolyte, while the interfacial resistance depends on the Li^+ transportation through the solid electrolyte interface (SEI). The elastic electrolyte

possesses a high ionic conductivity of $2 \times 10^{-3} \text{ S cm}^{-1}$ at ambient temperature. Therefore, the bulk resistance of the Li symmetric cell is as low as 11.6 ohm cm^2 . In contrary, the SEI consists of decomposition products of the electrolyte, which generally have much lower ionic conductivities than the bulk electrolyte. That means it is much more difficult for Li^+ to pass through the SEI than transport in the bulk electrolyte. As a result, the interfacial resistance is larger than the bulk resistance, which is a common phenomenon in Li symmetrical cells^{23, 24}. Fortunately, SEI between Li and the elastic electrolyte becomes stable after 50 hours (Supplementary Fig.14) and hinders further progress of the interface reaction, so that the Li symmetrical cell displays a long-term stability for 1800 hours.

5. In Fig. 5a, the cycle performance of $\mu\text{m-Si}$ half-cell was evaluated at 0.4 C with a lower cutoff voltage of 50 mV after the initial cycles, demonstrating a discharge capacity of 1039 mAh g^{-1} . However, a higher capacity of $\sim 2000 \text{ mAh g}^{-1}$ can be expected with a lower cutoff voltage of 50mV and low overpotential. How would the cycle performance be affected by a reduced C-rate and a higher discharge capacity, considering more severe expansion and shrinkage of Si particle?

Response: We thank the reviewer for this valuable comment. **We have conducted galvanostatic discharge-charge experiment with a reduced C-rate on the $\mu\text{m-Si}$ /elastic electrolyte/Li cell without external pressure.** As illustrated in Fig. R9, the fabricated cell achieved a high reversible specific capacity of $2909.7 \text{ mAh g}^{-1}$ at 0.1C ($1\text{C}=3579 \text{ mA g}^{-1}$) and maintained $2543.2 \text{ mAh g}^{-1}$ after 35 cycles. This result indicates a good stability of the $\mu\text{m-Si}$ with the elastic electrolyte, even though it underwent more severe volume fluctuation.

Fig. R9. (a) The charge-discharge curve of the 1st, 2nd, 15th, and 25th cycle and (b) the cycling stability test of the $\mu\text{m-Si}$ /elastic electrolyte/Li cell at 0.1C without external stack pressure.

6. The computational details of the finite element method to generate Figure 5b is not provided. Please add.

Response: We thank the reviewer for this thoughtful reminder. **We have provided the computational details of the finite element simulations in Method section; please see page 17 in the revised manuscript.** It has been highlighted in yellow in the revised manuscript and copied here: “The stress distribution and evolution in the $\mu\text{m-Si}$ electrode with $\text{Li}_6\text{PS}_5\text{Cl}$ and the elastic electrolyte were simulated through the finite element method. Firstly, models with randomly distributed Si spheres were established. The models were constructed based on the linear elasticity assumption, of which the relevant parameters of Si during lithiation can be found in the previous work⁵³. The implicit algorithm method was used and gradually iterated to the maximum expansion of Si to avoid convergence difficulties.”

7. What is the capacity ratio of LFP to $\mu\text{m-Si}$ in an LFP/ $\mu\text{m-Si}$ full cell? This information is crucial as the utilization rate of Si anodes is considered to have a significant impact on the cycle characteristics of full cells.

Response: We thank the reviewer for this insightful comment. The loading of LFP was approximately 6 mg cm^{-2} . Therefore, the capacity ratio of $\mu\text{m-Si}$ to LFP was 2.4 based on their theoretical specific capacities (170 mAh g^{-1} for LFP and 3579 mAh g^{-1} for $\mu\text{m-Si}$). **According to the reviewer's suggestion, we have information about the loading of LFP in the "Cell assembly and Electrochemical tests" part in Methods; please see page 16-17 in the revised manuscript.** It has been highlighted in yellow in the revised manuscript and copied here: "*As for the coin-type $\mu\text{m-Si}$ /elastic electrolyte/LFP cells, LFP cathode was prepared by mixing LFP, acetylene black and PVDF in a weight ratio of 8:1:1 in N-methylpyrrolidone. The homogenous slurry was coated on the Al foil by doctor blade with the LFP loading of around 6 mg cm^{-2} .*"

Reviewer #3:

The need for external pressure is a major obstacle to the application of all-solid-state batteries in practice, but the authors raised an effective strategy to alleviate this issue, with convincing demonstration using the micrometer-Si anode. The reviewer considers it as a highly important contribution to the all-solid-state battery community. However, before it can be accepted for publication, the following issues need further clarification.

Response: We would like to appreciate the reviewer for recognizing and highly praising the importance of our work.

1. The authors indicate that the synthesis of the elastic polymer electrolyte is conducted in Ar-filled glovebox. Is it unstable in ambient air? Which component in air will react with this material? Is it possible to improve its air stability in future studies? This directly influences the production cost, which is also rather important to the successful commercialization of all-solid-state batteries.

Response: We thank the reviewer for this comment. The elastic electrolyte is stable in dry air. In fact, the preparation of $\mu\text{m-Si}$ /elastic electrolyte/LFP pouch cells as well as the measurements of mechanical properties of the elastic electrolyte were completed in the dry room with a dew point of -40°C . No decomposition or performance degradation occurred to the elastic electrolyte after storing in the dry room. However, in ambient air where contains much moisture, the elastic electrolyte tends to absorb water due to the hydrophilicity of the copolymer network and LiFSI. The narrow electrochemical stability window of H_2O and its high reactivity toward electrodes and may lead to the degradation of batteries²⁵. This is a common issue confronting the lithium-ion battery electrolytes²⁶. Fortunately, dry rooms have been extensively used in the lithium-ion battery manufacturing industry and the cost has been controlled effectively. In future studies, constructing hydrophobic interphase on the electrode materials to improve their stability toward moisture²⁷ or adopting moisture-tolerating component such as ionic liquids in the electrolyte²⁵ can be potential solutions.

2. The authors claim that their elastic polymer electrolyte shows an oxidation potential of 4.5 V vs. Li/Li^+ , but the cell for cycling tests utilizes a 3 V-class material, LiFePO_4 , as the cathode. The reviewer would like to see the cycling performance of the cell where the elastic polymer electrolyte is paired with the 4 V-class cathodes like LiCoO_2 or $\text{LiNi}_{0.8}\text{Mn}_{0.1}\text{Co}_{0.1}\text{O}_2$.

Response: We thank the reviewer for this insightful comment. **We have applied the elastic electrolyte in the NMC cathode and conducted galvanostatic cycle tests on the NMC/elastic electrolyte/Li cells without external pressure.** To fabricate the NMC electrode with the elastic electrolyte, the precursor solution of the elastic electrolyte was dripped onto the NMC cathode coated on the Al foil, followed by a vacuum infiltration process and then UV polymerization. As illustrated in Fig. R10, the cell possessed a specific capacity of 208.6 mAh g^{-1} with an initial coulombic efficiency of 82.6% and could function normally for dozens of cycles, but the cycle stability still needed further improvements. It can be seen from Fig. R11 that NMC particles retained in tight contact with the elastic electrolyte after cycles, which ensured the rapid Li^+ transport inside the cathode. Consequently, it is reasonable to infer that the capacity decay was not caused by contact failure between the electrolyte and active materials, even though the cells were cycled with the exemption of external pressure.

Fig. R10. (a) The charge-discharge curve of the 1st, 5th, and 10th cycle and (b) the cycling stability test of the NMC/elastic electrolyte/Li cell without external pressure.

Fig. R11. Scanning electron microscopy images of the NMC electrode with the elastic electrolyte after cycles.

The capacity decay may be caused by the fact that the charge cutoff voltage of the cell (4.4 V versus Li⁺/Li) was quite close to the upper limit of the electrochemical stability window of the elastic electrolyte (4.5 V versus Li⁺/Li). In order to further enhance the electrochemical performances of the NMC/elastic electrolyte/Li cell, we employed 1 wt% LiPO₂F₂ as the additive in the elastic electrolyte in the cathode to construct a more stable cathode electrolyte interphase (CEI)⁷. The modified NMC/elastic electrolyte/Li cell working under no external stack pressure delivered a specific discharge capacity of 207.1 mAh g⁻¹ with an increased initial coulombic efficiency of 85.4%. Moreover, the cycle stability was meliorated and the cell maintained a capacity retention of 77.4% after 90 cycles (Fig. R12). The improved electrochemical performances were attributed to formation of the more robust CEI. It is reasonable to speculate that optimizing the type and amount of the additive could further enhance the compatibility of the elastic electrolyte with NMC. Whereas, this work is meant to focus on using the elastic electrolyte to tackle the mechanical failure issue of batteries working without high stack pressure. Further optimization of the electrochemical compatibility of the elastic electrolyte with NMC cathode remains a future work.

Fig. R12. (a) The charge-discharge curve of the 1st, 5th, and 10th cycle and (b) the cycling stability test of the Li/elastic electrolyte/NMC cell with 1 wt% LiPO₂F₂ as the additive in the cathode tested without external stack pressure.

3. The authors specified the areal mass loading of micrometer-Si in their cells, but the areal capacity (in mAh cm⁻²) that can be actually achieved in the cell is more meaningful. The desired mechanical properties reported here could also be helpful in reaching higher areal capacity. Therefore, the reviewer suggests the authors adjust the mass loading of micrometer-Si and investigate the maximum areal capacity that the micrometer-Si anode can achieve in the $\mu\text{m-Si/elastic electrolyte/Li cell with decent cycling stability.$

Response: We thank the reviewer for this comment. **In response to the suggestion from the reviewer, we conduct galvanostatic discharge and charge tests on solid-state $\mu\text{m-Si}$ with higher loading of 1.3 mg cm⁻² under no external stack pressure.** The $\mu\text{m-Si/elastic electrolyte/Li}$ cell was tested at 0.2C and 0.3C (1C=3579 mA g⁻¹). As illustrated in Fig. R13, the cell delivered an initial discharge capacity of 1377.8 mAh g⁻¹ with the coulombic efficiency of 83% at 0.2C (i.e. 0.9 mA cm⁻²). With an elevated current density to 0.3C (i.e. 1.4 mA cm⁻²), the reversible specific capacity decreased to 757 mAh g⁻¹. The reduced specific capacity may be caused by the enlarged overpotential of the Li counter electrode. It can be seen from the rate performance of the Li/elastic electrolyte/Li symmetric cell that the overpotential increased with the current density (Fig. R14). After 50 cycles, the $\mu\text{m-Si/elastic electrolyte/Li}$ cell maintained 631.3 mAh g⁻¹, corresponding to a capacity retention of 83.4%. Although the cycle life of the $\mu\text{m-Si/elastic electrolyte/Li}$ cell is curtailed with the increase of mass loading, it can be seen that the proposed elastic electrolyte remains effective to enhance the stability of the high-loading solid-state $\mu\text{m-Si}$ electrode operating without external pressure.

Fig. R13. The galvanostatic discharge and charge test on the $\mu\text{m-Si/elastic electrolyte/Li}$ cell with a $\mu\text{m-Si}$ loading of 1.3 mg cm⁻² at 0.2C and 0.3C without external stack pressure.

Fig. R14. Rate performance of the Li/elastic electrolyte/Li symmetric cell with an areal capacity of 0.1 mAh cm⁻² under no external stack pressure.

References:

1. Hansen, B.B. et al. Deep Eutectic Solvents: A Review of Fundamentals and Applications. *Chem Rev* **121**, 1232-1285 (2021).
2. Li, C.L. et al. A Low-Volatile and Durable Deep Eutectic Electrolyte for High-Performance Lithium-Oxygen Battery. *J Am Chem Soc* **144**, 5827-5833 (2022).
3. Zhou, D., Shanmukaraj, D., Tkacheva, A., Armand, M. & Wang, G.X. Polymer Electrolytes for Lithium-Based Batteries: Advances and Prospects. *Chem-Us* **5**, 2326-2352 (2019).
4. Lee, J.H., Kim, S., Cho, M., Chanthad, C. & Lee, Y. Crosslinked Gel Polymer Electrolytes for Si Anodes in Li-Ion Batteries. *J Electrochem Soc* **166**, A2755-A2761 (2019).
5. Bok, T. et al. An effective coupling of nanostructured Si and gel polymer electrolytes for high-performance lithium-ion battery anodes. *RCS Advances* **6**, 6960-6966 (2016).
6. Jia, H.P. et al. A novel approach to synthesize micrometer-sized porous silicon as a high performance anode for lithium-ion batteries. *Nano Energy* **50**, 589-597 (2018).
7. Tan, S. et al. Additive engineering for robust interphases to stabilize high-Ni layered structures at ultra-high voltage of 4.8 V. *Nature Energy* **7**, 484-494 (2022).
8. Yagci, Y., Jockusch, S. & Turro, N.J. Photoinitiated Polymerization: Advances, Challenges, and Opportunities. *Macromolecules* **43**, 6245-6260 (2010).
9. Zhong, R., Hu, H. & Zhou, Y.F. Synthesis and Characterization of a Trifunctional Photoinitiator Based on Two Commercial Photoinitiators with α -Hydroxyl Ketone Structure. *Materials* **14** (2021).
10. Su, Y. et al. Rational design of a topological polymeric solid electrolyte for high-performance all-solid-state alkali metal batteries. *Nature Communications* **13** (2022).
11. Zeng, D. et al. Promoting favorable interfacial properties in lithium-based batteries using chlorine-rich sulfide inorganic solid-state electrolytes. *Nat Commun* **13**, 1909 (2022).
12. Huang, Q. et al. Supremely elastic gel polymer electrolyte enables a reliable electrode structure for silicon-based anodes. *Nat Commun* **10** (2019).
13. Prakash, P. et al. A soft co-crystalline solid electrolyte for lithium-ion batteries. *Nat Mater* (2023).
14. Dai, T. et al. Inorganic glass electrolytes with polymer-like viscoelasticity. *Nature Energy* **8**, 1221-1228 (2023).
15. Veith, G.M. et al. Determination of the Solid Electrolyte Interphase Structure Grown on a Silicon Electrode Using a Fluoroethylene Carbonate Additive. *Sci Rep-Uk* **7** (2017).
16. Xu, C. et al. Improved Performance of the Silicon Anode for Li-Ion Batteries: Understanding the Surface Modification Mechanism of Fluoroethylene Carbonate as an Effective Electrolyte Additive. *Chem Mater* **27**, 2591-2599 (2015).
17. Dalavi, S., Guduru, P. & Lucht, B.L. Performance Enhancing Electrolyte Additives for Lithium Ion Batteries with Silicon Anodes. *J Electrochem Soc* **159**, A642-A646 (2012).
18. Schroder, K. et al. The Effect of Fluoroethylene Carbonate as an Additive on the Solid Electrolyte Interphase on Silicon Lithium-Ion Electrodes. *Chem Mater* **27**, 5531-5542 (2015).
19. Jin, Y.T. et al. Understanding Fluoroethylene Carbonate and Vinylene Carbonate Based Electrolytes for Si Anodes in Lithium Ion Batteries with NMR Spectroscopy. *J Am Chem Soc* **140**, 9854-9867 (2018).
20. Etacheri, V. et al. Effect of Fluoroethylene Carbonate (FEC) on the Performance and Surface Chemistry of Si-Nanowire Li-Ion Battery Anodes. *Langmuir* **28**, 965-976 (2012).

21. McCormick, C.L. & Chen, G.S. Water-Soluble Copolymers . IX. Copolymers of Acrylamide with N-(1,1-Dimethyl-3-Oxybutyl)Acrylamide and N,N-Dimethylacrylamide: Synthesis and Characterization. *J Polym Sci Pol Chem* **22**, 3633-3647 (1984).
22. Beckingham, B.S., Sanoja, G.E. & Lynd, N.A. Simple and Accurate Determination of Reactivity Ratios Using a Nonterminal Model of Chain Copolymerization. *Macromolecules* **48**, 6922-6930 (2015).
23. Lee, M.J. et al. Elastomeric electrolytes for high-energy solid-state lithium batteries. *Nature* **601**, 217-222 (2022).
24. Liu, M. et al. Improving Li-ion interfacial transport in hybrid solid electrolytes. *Nat Nanotechnol* (2022).
25. Liu, Q., Jiang, W., Yang, Z.Z. & Zhang, Z.C. An Environmentally Benign Electrolyte for High Energy Lithium Metal Batteries. *Acs Appl Mater Inter* **13**, 58229-58237 (2021).
26. Lux, S.F. et al. The mechanism of HF formation in LiPF₆ based organic carbonate electrolytes. *Electrochem Commun* **14**, 47-50 (2012).
27. Shen, X.W. et al. Lithium anode stable in air for low-cost fabrication of a dendrite-free lithium battery. *Nat Commun* **10** (2019).

REVIEWER COMMENTS

Reviewer #1 (Remarks to the Author):

I appreciate the specific replies from the authors. Overall, the addition of liquid components including a deep eutectic mixture and FEC additives lead me to view this electrolyte as a gel, rather than solid-state polymer. My concerns are not well addressed regarding this.

-General question 1: "Firstly, the electrolyte seems to resemble a gel polymer electrolyte rather than a solid electrolyte since a liquid solution (N-methylacetamide-LIFSI) is used. With the presence of the liquid phase, high pressure is not necessary to maintain good ionic conduction. Poly(dimethylacrylamide)-based electrolytes have been extensively studied in the battery field (Macromolecules 1996, 29, 1, 143– 155; Solid State Ionics 2003, 157, 233-239; Electrochimica Acta 1995, 40, 2417-2420)."

I agree with the authors about the understanding of the deep eutectic mixture. However, it is still liquid rather than solid, right? I am afraid that my question has not been addressed.

-General question 2: Is the low specific capacity attributed to poor electron or ion transfer? Can the polymer electrolyte reported in this work lead to a higher capacity utilization of NMC?

-Fig. 2b Since no crosslinking was seen, the electrolyte system is composed of a polymer and deep eutectic mixture. I prefer gel instead of solid-state.

Reviewer #2 (Remarks to the Author):

I think the authors have replied to all my questions. I am happy to suggest accepting it.

Reviewer #3 (Remarks to the Author):

Although the authors have satisfactorily addressed the concerns in the response letter, none of the data or points they made is added in the revised manuscript accordingly. The reviewer suggests that the authors do so, to help the readers gain a more comprehensive understanding on the material reported here.

Responses to Reviewers' Comments

We sincerely thank all reviewers for their constructive comments and suggestions. We have modified the manuscript accordingly, and listed the detailed corrections below point by point for each reviewer. All revised portions have been marked in yellow in the revised manuscript. The main corrections and the responses to the reviewers' comments are as follows.

Point-to-Point Responses

Reviewer #1:

I appreciate the specific replies from the authors. Overall, the addition of liquid components including a deep eutectic mixture and FEC additives lead me to view this electrolyte as a gel, rather than solid-state polymer. My concerns are not well addressed regarding this.

General question 1: "Firstly, the electrolyte seems to resemble a gel polymer electrolyte rather than a solid electrolyte since a liquid solution (N-methylacetamide-LIFSI) is used. With the presence of the liquid phase, high pressure is not necessary to maintain good ionic conduction. Poly(dimethylacrylamide)-based electrolytes have been extensively studied in the battery field (Macromolecules 1996, 29, 1, 143–155; Solid State Ionics 2003, 157, 233-239; Electrochimica Acta 1995, 40, 2417-2420)."

I agree with the authors about the understanding of the deep eutectic mixture. However, it is still liquid rather than solid, right? I am afraid that my question has not been addressed.

Response: Thanks the reviewer for providing such professional evaluations. We have thoroughly understood the reviewer's comments. I apologize for not clearly articulating our academic ideas. Allow me to reintroduce our work on the elastic electrolyte.

The reviewer is correct in stating that conventional polymer electrolytes mixed with liquid organic solvents can be considered gel polymer electrolytes. For instance, the common combination of PVDF polymer and dimethylformamide (DMF) solvent can be viewed as a gel polymer electrolyte. This electrolyte contains a certain amount of free solvent, exhibiting characteristics similar to those of the liquid solvent (DMF), such as volatility, low boiling point, and easy interfacial wetting.

As shown in Figure R1, thermal gravimetric data reveals that the temperature range for the evaporation of free DMF solvent is from 55.6 °C to 110 °C. Correspondingly, the temperature range for solvent evaporation in the PVDF gel containing DMF is from 57.1 °C to 109 °C. The difference between the two is minimal, indicating that there is no interaction between PVDF polymer and DMF liquid solvent; it is a simple physical mixture. This scenario is what we commonly refer to as a semi-solid electrolyte or gel electrolyte.

Fig. R1. Thermogravimetric analysis (TGA) curves of PVDF with DMF residue and pure DMF.

However, our work in this paper differs significantly from the above situation. N-methylacetamide (NMA), serving as the solvent, is initially solid at room temperature and liquifies when mixed with LiFSI to form a deep eutectic material (DEM). This DEM is indeed liquid in its free state. Still, after binding with the polymer formed by copolymerization with dimethyl acrylamide (DMAM) and acrylamide (AM), it is constrained and coordinated by the large molecular chains inside the polymer. The surrounding may slightly deviate the DEM from its deep eutectic point. This interaction renders the deep eutectic electrolyte no longer possessing the physical properties of free solvent molecules, but rather exhibiting a tendency towards solid-state characteristics. This interaction also imparts excellent mechanical properties to the elastic electrolyte mentioned in this work. Evidence for this phenomenon can still be obtained through thermal gravimetric experiments on the material.

Fig. R2. Thermogravimetric analysis (TGA) curves of the elastic electrolyte and the NMA-LiFSI eutectic mixture.

We can observe the solvent evaporation characteristics of the free-state and polymer-bound DEM through thermal gravimetric experiments. As shown in Figure R2, the NMA solvent in the free-state deep eutectic electrolyte evaporates in the temperature range of 82.2 to 220 °C. The LiFSI salt undergoes decomposition between 255 and 325 °C. However, when the deep eutectic electrolyte is combined with the specific polymer electrolyte to form the elastic electrolyte reported in this work, all transition temperatures experience a significant increase. Below 200 °C, there is almost no liquid evaporation. NMA solvent evaporation occurs only when the temperature rises to the range of 204

to 320 °C. The decomposition temperature of LiFSI salt also increases to the range of 325 to 400 °C. The significant increases in the evaporation temperature of NMA and decomposition temperature of LiFSI can be attributed to the effects of confinement and coordination of the eutectic mixture inside the copolymer framework, which makes it distinct from conventional free-state liquid additives in gel electrolytes.

The phenomenon of solidification characteristics observed in this deep eutectic electrolyte after interacting with specific polymers is a significant discovery emphasized in this paper. Based on this phenomenon, the elastic electrolyte also exhibits excellent mechanical properties. The mass ratio of the solvent N-methylacetamide (NMA) in the practical elastic electrolyte is close to 40%. However, contrary to conventional definitions that consider any addition of liquid solvent to a polymer as constituting a gel electrolyte, and relying on the traditional approach of defining the degree of solidification based on the amount of added liquid electrolyte, we believe these conventional definitions are not suitable for our work. Therefore, this elastic electrolyte can still be considered a solid-state electrolyte. Of course, we also do not refer to it as an all-solid-state electrolyte.

General question 2: Is the low specific capacity attributed to poor electron or ion transfer? Can the polymer electrolyte reported in this work lead to a higher capacity utilization of NMC?

Response: Thank you for this comment. In the mentioned reference (*Science* **2021**, 373, 1494-1499), the specific capacity of the NMC811 is calculated to be only ~80 mAh g⁻¹ in the μ m-Si/NMC cell. It is reasonable to attribute the low specific capacity to the sluggish ion transfer inside the cathode for the following reasons. Firstly, the poor point-to-point contact between Li₆PS₅Cl electrolyte particles and NMC811 may fail to provide sufficient ion transport sites. In addition, the high Young's modulus of the electrolyte (20-30 GPa for Li₆PS₅Cl, cited from *J Power Sources* **483** (2021)) makes it difficult to deform and create intimate interfacial contact with NMC811. By contrast, the elastic electrolyte can provide sufficient ion transport inside the cathode, thus lead to a higher specific capacity of NMC811 (207.1 mAh g⁻¹ with an initial coulombic efficiency of 85.4%, as shown in **Fig. R3**). This is because that the precursor solution of the electrolyte can fully infiltrate the porous electrode by a vacuum infiltration process and the elastic electrolyte with high ionic conductivity tightly surrounds the active materials after UV polymerization. Furthermore, as shown in **Fig. R4**, NMC particles retained in compact contact with the elastic electrolyte after cycles, which ensured the rapid Li⁺ transport inside the cathode. In this way, the high capacity of the NMC811 can be fully released.

Fig. R3. (a) The charge-discharge curve of the 1st, 5th, and 10th cycle and (b) the cycling stability test of the Li/elastic electrolyte/NMC cell with 1 wt% LiPO₂F₂ as the additive in the cathode tested without external stack pressure.

Fig. R4. Scanning electron microscopy images of the NMC electrode with the elastic electrolyte after cycles.

Fig. 2b Since no crosslinking was seen, the electrolyte system is composed of a polymer and deep eutectic mixture. I prefer gel instead of solid-state.

Response: Thank you for your comment. The reasons we categorize the elastic electrolyte as solid-state (not all-solid-state) have been explained above. We sincerely hope that you are satisfied with the response.

Reviewer #2:

I think the authors have replied to all my questions. I am happy to suggest accepting it.

Response: Many thanks for the reviewer's recommendation for publication.

Reviewer #3:

Although the authors have satisfactorily addressed the concerns in the response letter, none of the data or points they made is added in the revised manuscript accordingly. The reviewer suggests that the authors do so, to help the readers gain a more comprehensive understanding on the material reported here.

Response: Thank you for your kind suggestion. We have added the electrochemical performance data and the relevant discussions on the $\mu\text{-Si}$ /elastic electrolyte/Li cell with $\mu\text{-Si}$ loading of 1.3 mg cm^{-2} and the NMC/elastic electrolyte/Li cells in the revised manuscript. We believe that this modification improves the integrity of our work and helps the readers gain a more comprehensive understanding. The revised portions have been marked in yellow; please see page 11-13, Supplementary Fig. 18 and Supplementary Fig. 22.

References:

1. Ohashi, A., Kodama, M., Horikawa, N. & Hirai, S. Effect of Young's modulus of active materials on ion transport through solid electrolyte in all-solid-state lithium-ion battery. *J Power Sources* **483** (2021).

REVIEWERS' COMMENTS

Reviewer #1 (Remarks to the Author):

I'd like to thank the authors for the detailed replies and constructive discussions. My questions have been well addressed. I am happy to recommend publication of this excellent work.